# Uncertainty-informed selection of CMIP6 Earth System Model subsets for use in multisectoral and impact models

Abigail Snyder[1], Noah Prime[1], Claudia Tebaldi[1], Kalyn Dorheim[1]

[1]Pacific Northwest National Laboratory, Joint Global Change Research Institute, College Park MD, 20740, USA

*Correspondence to*: Abigail Snyder (abigail.snyder@pnnl.gov)

**Abstract.** Earth System Models (ESMs) and General Circulation Models (GCMs) are heavily used to provide inputs to sectoral impact and multisector dynamic models, which include representations of energy, water, land, economics, and their interactions. Therefore, representing the full range of model uncertainty, scenario uncertainty, and interannual variability that ensembles of these models capture is critical to the exploration of the future co-evolution of the integrated human-Earth system. The pre-eminent source of these ensembles has been the Coupled Model Intercomparison Project (CMIP). With more modeling centers participating in each new CMIP phase, the size of the model archive is rapidly increasing, which can be intractable for impact modelers to effectively utilize due to computational constraints and the challenges of analyzing large datasets. In this work, we present a method to select a subset of the latest phase, CMIP6, models for use as inputs to a sectoral impact or multisector dynamics models, while prioritizing preservation of the range of model uncertainty, scenario uncertainty, and interannual variability of the full CMIP6 ensemble results. This method is intended to help impact modelers select climate information from the CMIP archive efficiently for use in downstream models that require global coverage of climate information. This is particularly critical for large ensemble experiments of multisector dynamic models that may be varying additional features beyond climate inputs in a factorial design, thus putting constraints on the number of climate simulations that can be used. We focus on temperature and precipitation outputs of CMIP6 models as these are two of the most used variables among impact models and many other key input variables for impacts are at least correlated with one or both of temperature and precipitation (e.g. relative humidity). Besides preserving the multi-model ensemble variance characteristics, we prioritize selecting CMIP6 models in the subset that preserve the very likely distribution of equilibrium climate sensitivity values as assessed by the latest IPCC report. This approach could be applied to other output variables of climate models and, possibly when combined with emulators, offers a flexible framework for designing more efficient experiments on human-relevant climate impacts. It can also provide greater insight into the properties of existing CMIP6 models.

## 1 Introduction

The future evolution of the integrated human-Earth system is highly uncertain, but one common approach to begin addressing this uncertainty is to use outputs from a variety of computationally expensive, highly detailed process-based Earth System Models (ESMs) and General Circulation Models (GCMs) run under different scenarios. This approach has been facilitated by the Coupled Model Intercomparison Project (CMIP; Eyring et al. 2016), which has organized experiments that are standardized across modeling centers. Scenario simulations from CMIP, most recently through ScenarioMIP (O'Neill et al. 2016), are commonly used as inputs to downstream sectoral impact and multisector dynamic models, both by individual modeling efforts and by large, coordinated impact

modeling projects, like AgMIP or ISIMIP (e.g. Rosenzweig et al. 2013; Rosenzweig et al. 2014; Warszawski et al. 2014; Frieler et al. 2017). Using such multi-model ensembles captures the process

and structural uncertainties represented by sampling across ESM/GCMs, scenario uncertainty, and, to the extent that an ESM/GCM runs multiple initial condition ensemble members for a scenario, internal variability of the individual ESM (Hawkins and Sutton 2009; Hawkins and Sutton 2011; Lehner et al. 2020). These Earth system uncertainties can then be propagated through an impact model (perhaps after bias-correction (Lange 2019)) to understand possible human-relevant outcomes.

       From the Earth system modelers who produce climate data to the impact and multisector dynamic modelers who use it, each step in this process is computationally expensive. For Earth system modelers, variability across ESM/GCMs' projections of future climate variables can be significant (Hawkins and Sutton 2009; Hawkins and Sutton 2011; Lehner et al. 2020) and so the participation of multiple

modeling centers running multiple scenarios is critical to understanding the future of the Earth system. Further, statistical evaluation (Tebaldi et al. 2021) suggests that 20-25 initial condition ensemble members for each scenario an ESM/GCM provides are needed to estimate the forced component of extreme metrics related to daily temperature and precipitation, which are key inputs to many impacts models covering hydrological, agricultural, energy and other sectors. Fortunately, emulation of

ESM/GCM outputs to infill missing scenarios and enrich initial condition ensembles continues to improve (Beusch, Gudmundsson, and Seneviratne 2020; Nath et al. 2022; Quilcaille et al. 2022; Tebaldi, Snyder, and Dorheim 2022). This suggests that ESM/GCMs don't necessarily have to provide all of the runs desired for capturing possible futures, but instead a subset of scenarios including initial condition ensembles for emulator training. The total burden across the modeling and analysis

community to sample across ESM/GCMs and scenarios still remains high, even with the potential efficiency provided by emulators. Downstream from the physical climate science community, impact modelers often seek to understand future climate impacts in the context of ESM uncertainty by using the outputs of multiple ESMs under multiple scenarios as inputs to impact models (e.g. (Prudhomme et al. 2014; Müller et al. 2021)). In a world unburdened by time and computing constraints, an impact

model would take as input every projected data set available (possibly weighted according to observation and/or by model independence) to have a full understanding of the total variance in possible outcomes. Our world includes those burdens, made even larger when impact models require bias-corrected climate data as input. This can quickly become an intractably-sized set of runs to perform and analyze for impact modelers.  For the multisector dynamics community, whose modelers often attempt

to integrate results from multiple impact models to understand interactions of different sectors (like energy, water, land, and economics) of the integrated human-Earth system (Graham et al. 2020) this challenge multiplies. Finally, multisector dynamic models are beginning to run large ensemble experiments that vary additional features beyond climate inputs in a factorial design (e.g. (Dolan et al. 2021, 2022; Guivarch et al. 2022)) further adding to the computational costs to be faced. The

multisector dynamics approach is the approach that the examples in this work focus on: downstream models that require global coverage of a variety of climate model output variables at different temporal scales. Were a study to be focused on particular regions or localized impacts and dynamics, other selection criteria, such as model skill (closeness to observation, ability to replicate modes of variability known to be particularly important to that region, etc.) and the effect of downscaling and bias

correction, known to introduce additional sources of variability and uncertainty (Lafferty and Sriver 2023) in that region could be explored.

For all communities involved, an efficient way to design and then use climate model runs is critical. While there is likely no perfect solution to balance the tension between the competing priorities of
different climate data creators and climate data users, this work describes a method for selecting a subset of CMIP6 models that prioritizes retaining the overall uncertainty characteristics of the entire data set, particularly in dimensions relevant to impact and multisectoral modelers. The method proposed here exists in the context of a rich literature on model selection, with methods focused on model skill in comparison to observation and/or tracking and controlling for climate model dependence (Abramowitz
et al. 2019; Brands 2022; Merrifield et al. 2023; Parding et al. 2020). These are critical aspects to consider when sub-selecting climate models for downstream use. Merrifield et al (2023) does include model spread as a critical consideration for model selection, but to our knowledge, there is no uncertainty-first consideration of climate model selection. The method we present in this work is an adaptable framework that could complement other approaches based on skill and climate model
independence, and some of the choices made in implementing this method may be adaptable for other uses or priorities.

## 2 Methods

We approach the question of uncertainty in the full collection of CMIP6 models as being one of understanding the total variance in the CMIP6 outputs, which, following the Hawkins and Sutton
framing of the problem (Hawkins and Sutton 2009; Hawkins and Sutton 2011; Lehner et al. 2020), we understand as coming from three sources: internal variability, scenario and model uncertainties. Rather than attributing fractions of total variance to different sources and optimizing that as part of the selection process, however, we focus on projecting the data into a new coordinate basis designed to maximize total variance. Principal Component Analysis (PCA) does exactly this: it identifies a new set
of basis vectors maximizing total variance that data can be projected into. Once climate model data has been projected into this space (e.g. as in Figure 3), it's possible to sample a subset of climate models that cover the spread of the projections of the full set of climate model outputs in this variance-maximizing space.

The overall steps of this method are summarized in Table 1. Sections 2.1 and 2.2 provide fuller details on using PCA to characterize the full set of climate model data (2.1) and selecting a representative subset of climate models within that characterization (2.2). Table 1 especially highlights the choices made for this particular effort, based on the authors' experience with multisectoral impact modeling. Section 2.3 outlines our approach to evaluating the extent to which our model subset preserves the
uncertainty properties of the full data set. Nothing in the method prevents its being adapted with different regions of interest, indices of behavior, or ESM/GCM output variables, although evaluation of results in new implementations would be necessary.

**Table 1. Summary of method**

| Step | Description | Experiment 1 | Experiment 2 |
|------|-------------|--------------|--------------|
| 1 | Identify relevant climate model output variables | Temperature, precipitation from all (22) ScenarioMIP Tier 1-participating models | Temperature, precipitation from independent* (16) ScenarioMIP Tier 1-participating models |
| 2 | Aggregate gridded time series to region-levels | IPCC WG1 non-arctic land | IPCC WG1 non-arctic land |
| 3 | Identify and extract region indices for each variable, for each Model-Scenario to capture characteristics of uncertainty of interest | Ensemble averaged: mid-century anomaly, end of century anomaly, interannual standard deviation | Ensemble averaged: mid-century anomaly, end of century anomaly, interannual standard deviation |
| 4 | Form a matrix of Model*Scenario rows and Region*Indices columns for the full data and perform PCA; identify number of eigenvectors, $N$, responsible for majority of variance | $N=5$ eigenvectors | $N=5$ eigenvectors |
| 5 | Create candidate subsets of models based on heuristic filters of interest | Model subset size = 5; heuristic filter is that each subset must preserve the IPCC distribution of equilibrium climate sensitivity. | Model subset size = 5; heuristic filter is that each subset must preserve the IPCC distribution of equilibrium climate sensitivity. |
| 6 | Calculate the summary metric for each subset and select the subset with the smallest value | Minimize distance from out-of-subset model to a subset model across the $N=5$ dimensions. | Minimize distance from out-of-subset model to a subset model across the $N=5$ dimensions. |
| 7 | Calculate the Hawkins and Sutton partitions for the full set of data and selected subset and use as independent, qualitative evaluation data | Full data = 22 models Subset = ACCESS-CM2, ACCESS-ESM1-5, CMCC-ESM2, MRI-ESM2-0, GFDL-ESM4 | Full data = 16 models Subset = IPSL-CM6A-LR, ACCESS-ESM1-5, MRI-ESM2-0, BCC-CSM2-MR, MIROC6 |
| | * independent as defined in this work, many definitions exist | | |

## 2.1 Data preparation and characterization

Impact models often require multiple output variables from a climate model on daily or monthly time scales, with temperature and precipitation being the most common variables needed. For tractability, we focus on the IPCC WG1 non-arctic land regions (Iturbide et al. 2022), as these regions are primarily
where humans live, consume water, generate electricity, and grow food. I.e., the places most relevant in multisectoral models of the integrated human-Earth system. We also limit ourselves to ESM/GCMs that completed all four ScenarioMIP Tier 1 experiments (Table 2). This still results in more than 600 trajectories across models, scenarios, and ensemble members for each region.

In this work, we are treating this collection of ESM/GCMs and scenario results in these regions as the full set of data of which we would like to faithfully represent the uncertainty characteristics, and then select a subset of climate models for impact modelers to use, based on preserving those characteristics. Critically, however, is that once the full set of climate data is characterized, as we outline in this section, the selection step of the method includes a step to restrict the ECS distribution of
the model subset to reflect that of the IPCC AR6-defined most likely distribution (Core Writing Team, H. Lee and J. Romero (eds. ) 2023). This shifts the average ECS value of the selected subset down relative to the existing full data covered in Table 2. Following this ECS distribution, a single high ECS climate model is allowed to be included in the subset, allowing both the 'hot model problem' (Hausfather et al. 2022) to be addressed as part of the model subset selection process as well as ensuring
that a range of model behaviors across different ECS values are included. Models for which we could not readily identify ECS values in the literature are included in the characterization of the full space but they are not eligible for selection in the subset, as preserving the IPCC distribution of ECS values is a critical filter in this selection process for the examples outlined in this work (more details in Section 2.2).

**Table 2. Models and scenarios making up the full set of data, as well as their equilibrium climate sensitivity (ECS) values sourced from *(Meehl et al. 2020; Lovato et al. 2022; Zelinka et al. 2020)*. Note that even the Earth System Models in CMIP6 run these experiments in concentration-driven mode rather than emissions-driven mode.**

| ESM | ECS | SSP126 Ensemble size | SSP245 Ensemble size | SSP370 Ensemble size | SSP585 Ensemble size |
|---|---|---|---|---|---|
| ACCESS-CM2 | 4.7 | 5 | 5 | 5 | 5 |
| ACCESS-ESM1-5 | 3.9 | 40 | 10 | 30 | 40 |
| BCC-CSM2- | 3.0 | 1 | 1 | 1 | 1 |

| MR | | | | | |
|---|---|---|---|---|---|
| CAMS-CSM1-0 | 2.3 | 2 | 2 | 2 | 2 |
| CESM2 | 5.2 | 3 | 3 | 3 | 3 |
| CESM2-WACCM | 4.8 | 1 | 3 | 1 | 3 |
| CMCC-CM2-SR5 | 3.52 | 1 | 1 | 1 | 1 |
| CMCC-ESM2 | 3.57 | 1 | 1 | 1 | 1 |
| CanESM5 | 5.6 | 25 | 25 | 25 | 25 |
| EC-Earth3-Veg-LR | 4.2 | 3 | 3 | 3 | 3 |
| FGOALS-f3-L | 3.0 | 1 | 1 | 1 | 1 |
| FGOALS-g3 | 2.87 | 4 | 4 | 4 | 4 |
| GFDL-ESM4 | 2.6 | 1 | 3 | 1 | 1 |
| INM-CM4-8 | 1.8 | 1 | 1 | 1 | 1 |
| INM-CM5-0 | 1.9 | 1 | 1 | 5 | 1 |
| IPSL-CM6A-LR | 4.6 | 6 | 11 | 11 | 6 |
| MIROC6 | 2.6 | 50 | 33 | 3 | 50 |
| MPI-ESM1-2-HR | 3.0 | 2 | 2 | 10 | 2 |
| MPI-ESM1-2-LR | 3.0 | 10 | 10 | 10 | 10 |
| MRI-ESM2-0 | 3.2 | 5 | 5 | 5 | 5 |
| NorESM2-MM | 2.5 | 1 | 2 | 1 | 1 |
| UKESM1-0-LL | 5.3 | 13 | 14 | 13 | 5 |


For each scenario, region, and available ensemble member in each in each ESM/GCM, we extract the following temperature and precipitation outputs: mid-century (2040-2059) average anomaly relative to that model's historical average (1995-2014), the end of century (2080-2099) anomaly relative to historical average, and the interannual standard deviation (IASD). Interannual standard deviation is calculated by detrending the regional average level temperature and precipitation time series from 1994-2100 using non-parametric locally weighted smoothing (LOESS as implemented in the python statsmodels package), and then taking the standard deviation of the residuals. For each scenario and model, these ensemble-member values are used to calculate the ensemble average to form our final indices in each region. These six indices (three for each of temperature and precipitation) per ESM-scenario-region combination are selected to result in data that represents the model uncertainty, scenario uncertainty, and interannual variability of our full set of data. By focusing on ensemble averages, models that performed more realizations are not over-represented in the overall space. When an ensemble size is only one realization, that realization's value is used. The key question is how to efficiently characterize this collection of data in a way that enables an efficient subsampling of models that still preserves the main dimensions of variations of the full ensemble.

This full data can be written as a matrix $A$ with *Nmodel*Nscenarios* rows and *Nindices*Nregions* columns. In the case of considering all 22 models listed in Table 2 as representative of the full space, this is 88 rows and 258 columns and we use these numbers for simplicity in some of the vector descriptions that follow. Below, we outline two experiments that highlight the adaptability of this method by considering model dependency in the CMIP6 models versus not. In the case of restricting to independent models only to make up the full data, these numbers of course change.

Principal components analysis (PCA) is then a natural technique to understand the variance of this full data set by forming the covariance matrix $S = A^T A$. The eigenvectors of $S$ are a set of orthogonal basis vectors (each vector is length 258) that are ordered by how much variance of the full data each eigenvector explains. Mathematically, this means that each row of $A$, ($\{\vec{a_i}|i = 1 \dots 88\}$) representing the indices in all regions for a single climate model-scenario) can be projected into the space of eigenvectors $\{\vec{PC_i}|i = 1 \dots 88\}$ and written as $\vec{a_i} = \Sigma_j c_{ij} \vec{PC_j}$ for projection coefficients (coordinates in the basis of eigenvectors), $c_{ij}$. Thus $\vec{PC_1}$, for example represents some pattern of joint, spatiotemporal temperature and precipitation behaviors that explains the greatest variance across ESM/GCM-scenario observations. Each CMIP6 model-scenario combination has some contribution from this pattern described by its projection coefficient, $c_{i1}$. This projection can be done over all eigenvectors, or as is common with PCA, a small subset of the eigenvectors that explain the majority of variance.

To demonstrate the flexibility of this approach to characterizing data, we perform the same analysis in two different experiments:
- Experiment 1 assumes all 22 models listed in Table 2 make up the full data.

- Experiment 2 assumes the full data is made up of only 16 of the models in Table 2, with ACCESS-CM2, CESM2-WACCM, CMCC-CM2-SR5, FGOALS-f3-L, INM-CM4-8, and MPI-ESM1-2-HR being removed from consideration as they share clear model dependencies with other models in the full data. When deciding which of two related models to keep, we chose

based on keeping the model with greater number of realizations as this is valuable for downstream uses. Other criteria could be used to define model dependency and make selections, as determining model independence is itself a rich field of study (Abramowitz et al. 2019; Brands 2022; Merrifield et al. 2023).

Figure 1 is a plot of the fraction of variance explained by each of the first 15 eigenvectors in each experiment. Based on this figure, we restrict ourselves to the first five eigenvectors for projections (just after the 'elbow'), explaining more than 70% of total variance for each experiment. The number of eigenvectors considered is another area of flexibility of this method. There is no reason this method could not be applied to more or even all of the eigenvectors. However, the more eigenvectors that are

considered, the higher dimensional the space that model selection must take place in. This slows down the calculations for selecting a subset considerably, at the benefit of explaining only a few extra percent of total variance with each vector added.

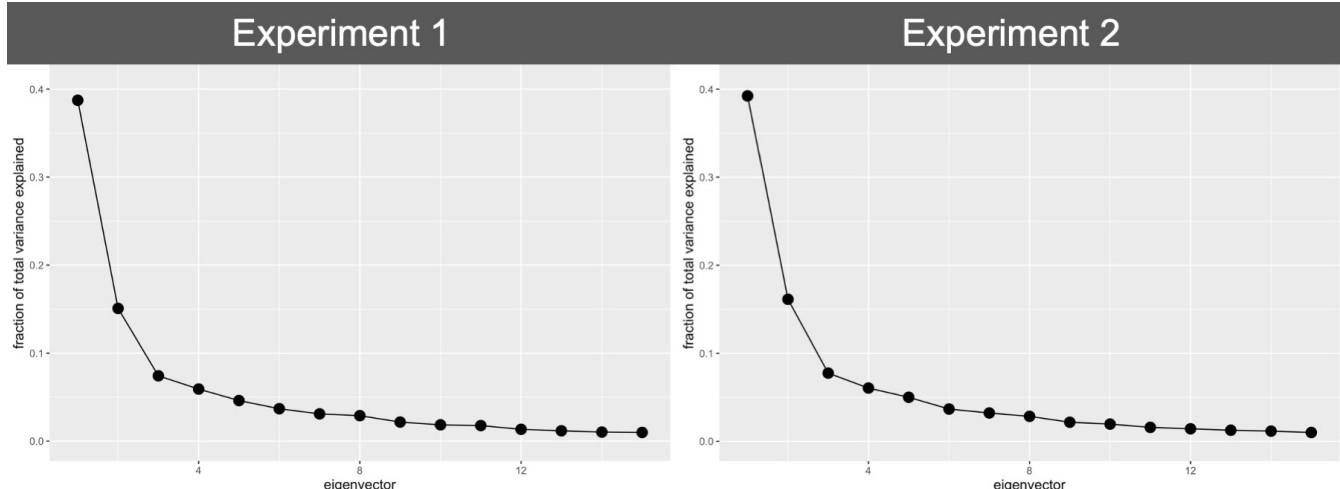

**Figure 1: fraction of variance explained by each eigenvector of the principal component analysis on scaled data for Experiment 1 (left) and Experiment 2 (right)), for the first 15 eigenvectors.**

Figure 2 is a visual representation of these five eigenvectors for each experiment. Each row is a map of all indices for each eigenvector. While it is tempting to interpret differences in sign as meaningful, note

that these are centered and scaled variables. Interpretation of the eigenvectors is also less meaningful than the fact that they represent an orthogonal coordinate system that maximizes total variance. For both experiments $\overrightarrow{PC_1}$ is dominated by temperature and, to a lesser extent, high latitude precipitation, highlighting that these features are responsible for 38.7% of the total variance of our full set of data (from Fig. 1). This is not the only contribution to total variance of temperature, of course, but it is a

good sanity check that temperature anomalies are the most dominant feature in the dimension explaining the highest fraction of total variance. $\overrightarrow{PC_2}$ is dominated by temperature interannual variability and high latitude precipitation interannual variability. $\overrightarrow{PC_3}$ to $\overrightarrow{PC_5}$ feature a mix of the indices, with strong emphasis on precipitation related behaviors. Note that because we treated temperature and precipitation indices together in one matrix, the eigenvectors include joint temperature-

precipitation behaviors that may be missed if the variables were treated separately. When comparing each map between the two experiments, it is worth noting that the spatial patterns are very similar between Experiment 1 and Experiment 2. Specifically, it is primarily in the southern latitudes in $\overrightarrow{PC_5}$ (explaining only ~5% of total variance in the full data in either experiment) that clear differences between the two experiments begin to emerge. This suggests that the patterns of total variance in

this data set are dominated by differences beyond those that might be captured in our definition of model dependence. For example, maybe different representations of ocean physics are playing a large role. Testing of this hypothesis is outside the scope of this method description work but highlights the potential value of characterizing an archive of CMIP data in this way. In Figures 1 and 3, we also see that the fraction of total variance explained by each eigenvector is similar across the two experiments.

Overall, this similarity when accounting for model dependence versus not is not entirely surprising. The full data set in Experiment 1, with all of the model dependencies it includes, does include over-representation of whatever physics (for example) that are used in the most ESM/GCMs. However, because PCA is focused on maximizing total variance, this over-representation does get mitigated to an extent.


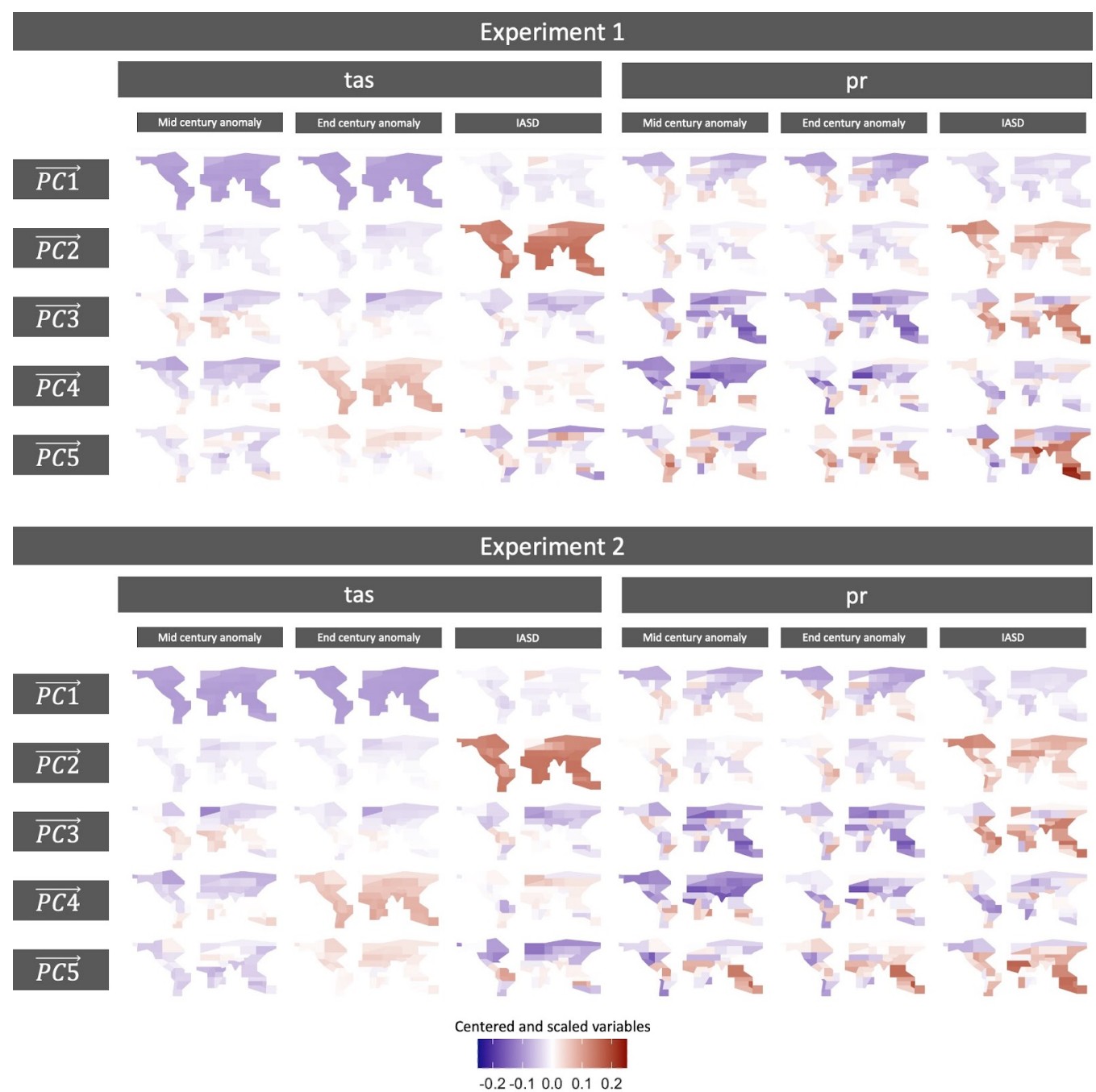

**Figure 2: Maps of the first five eigenvectors of our full data. Each row is a single eigenvector, with maps presented for each of the indices. Note that the colorbar scales are all standardized. A larger, landscape-oriented version of this figure is included in Appendix A (Fig. A1) for easier inspection.**

By treating the span of these five eigenvectors as the representative space of full data, we can project all data into this space and visualize its behavior by two-dimensional plots of all five PCs combinations. Figure 3 shows these 2-d slices of the projection coefficients for each ESM/GCM and scenario into this space for each experiment. These points in space are the $c_{ij}$ values in the principal component decomposition $\vec{a_i} = \Sigma_j c_{ij} \overrightarrow{PC_j}$, where $\vec{a_i}$ contains the indices in all regions for a single climate model-scenario. Because eigenvectors are orthogonal in PCA, together these panels are a complete visual representation of our ESM/GCM index data characterized in each $\vec{a_i}$, truncated to the first five projection dimensions (since they account for more than 70% of total variance in the full data in each experiment). If an impact modeler wished, they could run every model-scenario combination here for all available ensemble members. In practice, however, this may not be computationally tractable to either run or analyze. This view also motivates our approach for selecting our subset of climate models that preserve the uncertainty characteristics defined by this space. Because we want to represent the same characteristics of variance with fewer ESM/GCMs, our selection of a subset of ESM/GCMs is seeking to essentially sample this cloud at its extremes, middle, and throughout as subset size allows.

# Experiment 1

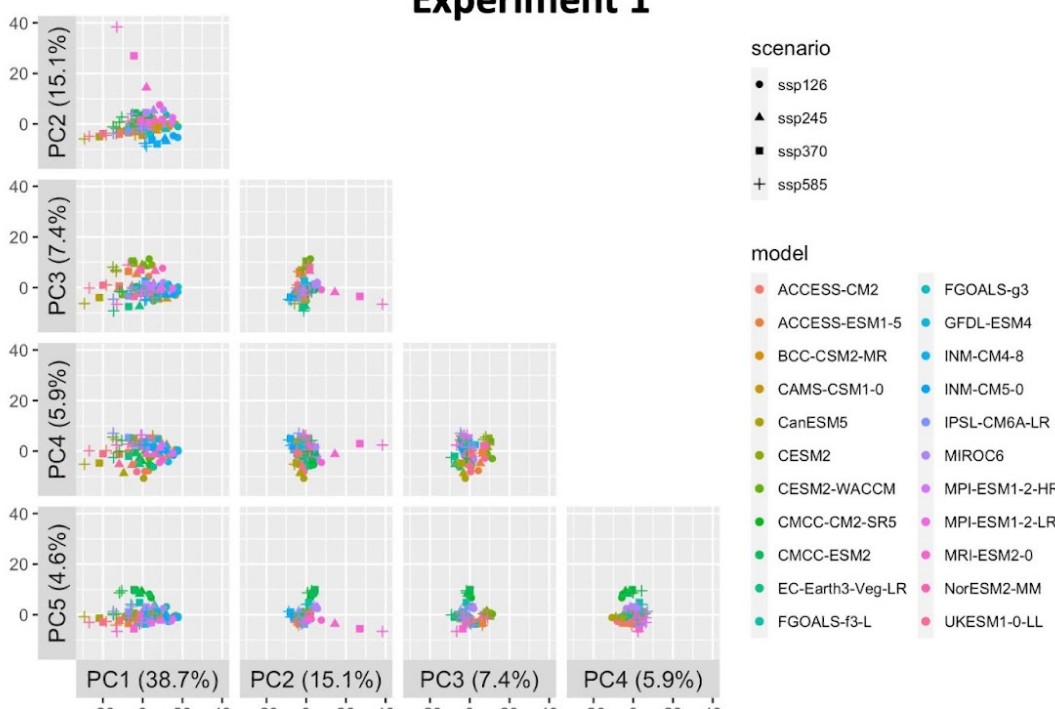

# Experiment 2

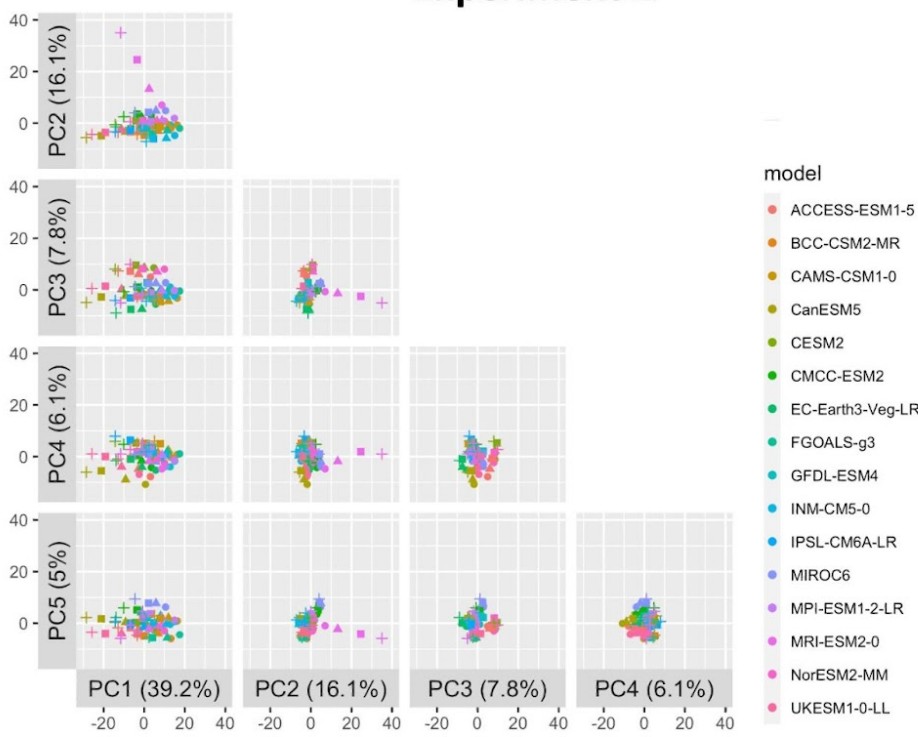

**Figure 3: 2-d slices of the projection coefficients for each ESM/GCM-scenario combination into the space spanned by the first five eigenvectors.**

## 2.2 Selection criteria of subset of CMIP6 models

Once the full set of data has been projected into the new basis identified to maximize total variance by PCA (as in Figure 3), selecting a representative subset of climate models across that space is relatively straightforward, and so is adding additional selection criteria, like constraining the distribution of ECS values. The subset of climate models that minimizes distance to all other climate models across this five-dimensional space is the subset selected. In more detail, first, subsets of candidate models are
formed (in this work, five models per subset, but the approach can be applied to any target subset size). While it would be possible to consider any combination of five models from the full set of 22, in this work we add a pre-filtering step. From all 22 choose 5 potential subsets, we only consider as candidate subsets the 150 subsets that roughly preserve the IPCC distribution of equilibrium climate sensitivity values and for which we could identify ECS values in the literature (Core Writing Team &
(eds.), 2023; Lovato et al., 2022; Meehl et al., 2020*; Zelinka et al. 2020*). Then for each subset, we step through each non-candidate model and calculate the minimum Euclidean distance to any of the subset's climate model's coefficients. The summary metric for each subset of candidates is then the average over all non-candidate model minimum distances, and the subset of candidate models with the smallest summary metric is the selected subset. Unlike many metrics (e.g. (Nash & Sutcliffe, 1970; Tebaldi et
al., 2020)), there is unfortunately not a clear threshold for 'good enough' performance based on this metric and so in the so in the next section, we provide a qualitative evaluation framework that assesses whether the selected subset is successful at preserving the major characteristics of the full ensemble's uncertainty characteristics.


## 2.3 Method for subset evaluation

The Hawkins and Sutton breakdown of total variance into relative sources of uncertainty inspired our choices of regional indices, both anomalies and interannual standard deviations. However, our subset selection is made in the space of the climate models' absolute positions, without formally considering
the relative breakdowns into fraction of total variance explained by model uncertainty, scenario uncertainty, and internal variability. Therefore, the partitioning of *relative* uncertainty calculated in the style of Hawkins and Sutton (Hawkins & Sutton, 2009, 2011) is a useful independent framework to evaluate the extent to which our climate model subset preserves the characteristics of the full ensemble. We don't expect perfect agreement in the Hawkins and Sutton (HS) fractions between our climate
model subset and the full data because we do change the distribution of ECS values in the subset we select. However, even qualitative discrepancies in the HS fractions between the full ensemble and the chosen subset can be useful to understand whether decisions such as constraining the distribution of ECS values are moving the relative contribution of each source of uncertainty in an explainable way.

The calculations of HS fractions are as follows: Consider a set of trajectories for a given climate variable produced by various model ESMs and scenarios. For example, this could be the annual average temperature or precipitation in a given world region. At each time step, $t$, there will be variation in the estimates from each observation in the set. The goal for a given set is to attribute a proportion of the variation or uncertainty at each time step to one of the three sources: interannual variation, model

uncertainty, and scenario uncertainty. In our application, we want to do this for a "full" model set and compare the distribution of assigned variance to the same analysis on a selected subset of models.

The crux of this method for separating uncertainty is to write the raw predictions for each observation as $X_{m,s,t} = x_{m,s,t} + i_{m,s} + \varepsilon_{m,s,t}$, where $X_{m,s,t}$ is the raw prediction for model $m$ scenario $s$ at time $t$, $x_{m,s,t}$

is a smoothed fit of the variable anomaly with reference period 1995-2014, $i_{m,s}$ is the average variable value over the reference period, and $\varepsilon_{m,s,t}$ is the residual, representing interannual variation (IAV). Note that while internal variability is itself a constant value for each climate model-scenario, the fraction of total variance that internal variability explains can change over time as the model and scenario components change. Similarly, while we do not want to select subsets of scenarios, understanding the

relative contribution of scenario uncertainty is critical to appreciate the variability across the different models.

We can then essentially calculate the interannual variation component as the variance of all $\varepsilon$, the model

uncertainty component at each time step as the variation in $x$ over the different models, and the scenario uncertainty at each time step as the variation in $x$ over the different scenarios. The variance calculations can have a weighting component, although in this work we treat all models included in each experiment-specific full ensemble as uniformly weighted. The interannual variability component is computed as $V = \sum_m var_{s,t}(\varepsilon_{m,s,t})$. The model uncertainty component is $M(t) =$

$\frac{1}{N_s}\sum_s var_m(x_{m,s,t})$ for the number of scenarios used $N_s$ (four in this study). The scenario uncertainty component is $S(t) = var_s(\sum_m x_{m,s,t})$.

Note that each of these components may, for example, be weighted based on each climate model's closeness to some observational set, but in this work we weight them uniformly, as we are not

concerned with model validation. Furthermore, following the assertion by Hawkins and Sutton (2009) assert that final fractions of total variability are not strongly affected by using different weights.

## 3 Results and discussion

The selected subset of ESM/GCMs and their respective ECS values are provided in Table 3 for each experiment. Figure 4 presents an identical plot to Fig. 3 but with the selected ESM/GCMs highlighted

by black box outlines to emphasize the extent to which the subset covers the full ensemble. We also perform a validation exercise based on the work of Hawkins and Sutton (Hawkins & Sutton, 2009, 2011) using the whole time series data rather than the 6 metrics that guided our subset selection to
provide an additional perspective on the ability of the method to preserve the characteristics of variability of the whole ensemble.

**Table 3. Selected Model subset and ECS values for each experiment. Models selected in both experiments in bold.**

| Experiment 1 Model (ECS value) | Experiment 2 Model (ECS value) |
|---|---|
| ACCESS-CM2 (4.70) | IPSL-CM6A-LR (4.6) |
| **ACCESS-ESM1-5 (3.9)** | **ACCESS-ESM1-5 (3.9)** |
| **MRI-ESM2-0 (3.2)** | **MRI-ESM2-0 (3.2)** |
| BCC-CSM2-MR (3.0) | MPI-ESM1-2-0 (3.0) |
| **MIROC6 (2.6)** | **MIROC6 (2.6)** |


# Experiment 1

# Experiment 2

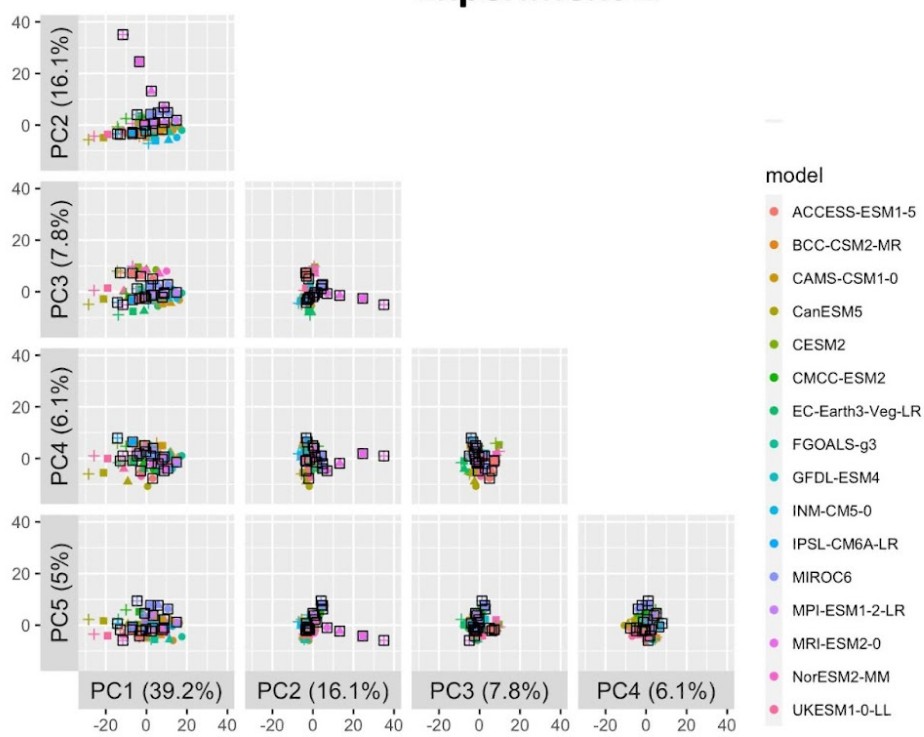

*Figure 4: Same as Figure 3 but with the selected ESM/GCMs highlighted by black box outlines or Experiment 1 (top) and Experiment 2 (bottom).*

## 3.1 Subset Evaluation

As noted in Section 2.3, the partitioning of total variance into the relative contribution of different sources calculated by Hawkins and Sutton (Hawkins & Sutton, 2009, 2011) is a useful independent framework to evaluate the extent to which our climate model subset preserves the characteristics of the full ensemble. As we did not calculate the specific time series of Hawkins and Sutton (HS) fractions for internal variability (there, as here, quantified as interannual variability after detrending the annual mean time series), scenario uncertainty, and model uncertainty to form any part of our selection procedure, we can use these HS fractions as independent evaluation criteria. We calculate the time series of HS fractions for temperature and precipitation separately in each region, for the full set of data and over just our selected subset of data, i.e., for each experiment, over the selection of CMIP6 models making up the full data set in that experiment, and only using the subset of 5 ESM/GCMs that our method identified. Details of these calculations are provided in Section 2.3. To manage the inspection of three time series for each of 86 region-variable combinations, we use root mean square error (RMSE) to compare the full data time series and the subset data time series from 2040 onward (as that is the focus of our indices) for each uncertainty partition, for each variable in each region.

Because of the large number of regions we wish to examine for two variables over time in each of two separate experiments, we seek some criteria to narrow this down. To identify specific region-variable combinations that are due for closer inspection, we set a threshold on the RMSE values for each uncertainty partition for each region-variable combination. As we note in Section 2.3, a discrepancy between the HS fractions for the subset and the full data is not a sign of poor selection. Rather, it merely means it is a region to inspect more closely and consider whether the discrepancies follow from our constraint of ECS values as part of our selection procedure. If any of the three uncertainty partitions have RMSE>0.1, we flag that region-variable combination for closer inspection. While thresholds like this are often arbitrary to set, each uncertainty partition for the subset data explaining the fraction of total variance within 10% of the full data's partition seems a good place to start. We show in Appendix A the results of a less stringent choice, namely, if we relax this to 20%, far fewer regions-variables get flagged for inspection in each experiment. Lowering this inspection threshold will of course flag more region-variables combinations, but as we point out below, a portion of the combinations flagged with a threshold of 0.1 still actually perform reasonably when plotted over time. Figure 5 provides a color-coded map of regions where temperature, precipitation, both, or neither have RMSE <= 0.1 for all three uncertainty partitions to give a sense of the spatial extent of performance.

## Experiment 1

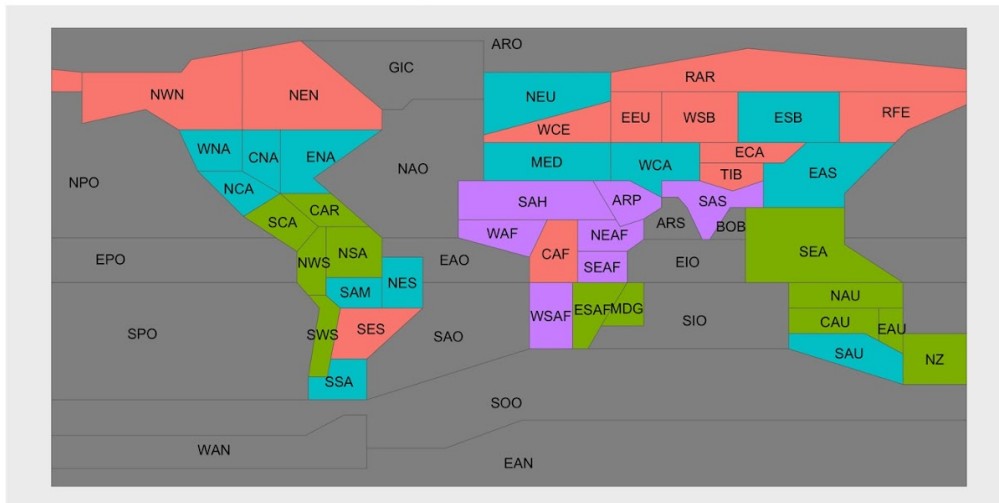

## Experiment 2

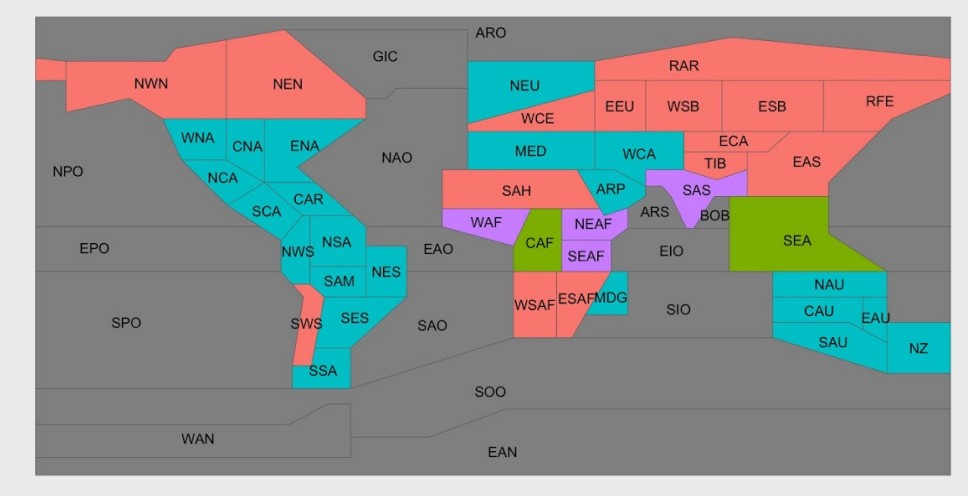

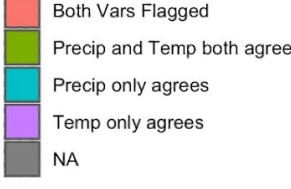

**Figure 5. a color-coded map of regions where temperature, precipitation, both, or neither have RMSE <= 0.1 for all three uncertainty partitions.**


The time series of HS fractions for the remaining region-variable combinations for which RMSE > 0.1 are plotted in Figure 6 (temperature) and Figure 7 (precipitation). For temperature in both experiments, we see that interannual variability is often performing well, with increasingly better performance over time. The partitioning of model and scenario uncertainty is where the subset's behavior begins to depart

from the full data, although this too tends to have smaller discrepancies as time goes on. This is not

surprising: in the full set of data, a good portion of model uncertainty is driven by different ECS values. As provided in Table 2, the values across ESM/GCMs that participated in Tier 1 ScenarioMIP experiments do not match the IPCC very likely distribution. By contrast, we are only selecting subsets of ESM/GCMs that match this distribution, overall resulting in a cooler collection of climate models

than the full data. This accounts for much of the discrepancy in the balance between scenario and model uncertainty contributions being different between our full and our subset data. Enforcing a different distribution of ECS values in the selected subset relative to the full data will also explain many of the discrepancies for precipitation, given the known strong correlation between temperature and precipitation changes. For precipitation, we overall see total uncertainty in the subset having a greater

fraction explained by interannual variability and less by model uncertainty across time. For both temperature and precipitation, the direction of these discrepancies is not surprising given our choice to reshape the distribution of ECS to an overall cooler collection than the full data. What we want to see in all panels of Figures 6 and 7, is a qualitative agreement with the relevance of the three sources of uncertainty in the full ensemble. We note that even in the regions we have flagged for closer inspection

in Figures 6 and 7, model uncertainty is evolving in the subset in much the same way it evolves in the full set, albeit with a shift. According to this criterion, most of the regions flagged by the application of the 0.1 threshold remain consistent with the full ensemble representation of the three uncertainty sources, for both variables and across both experiments. A small portion of the regions inspected in Figure 6 and 7 do ultimately simply differ more dramatically in the representations in the full set versus

subset, such as TIB in Experiment 1 in Figure 6. This is often unavoidable in a few regions when seeking to represent the entire globe with a subset of ESM/GCMs, again noting that even more substantial quality discrepancy such as this isn't a sign of failure of the method due to the constraints on ECS distributions.


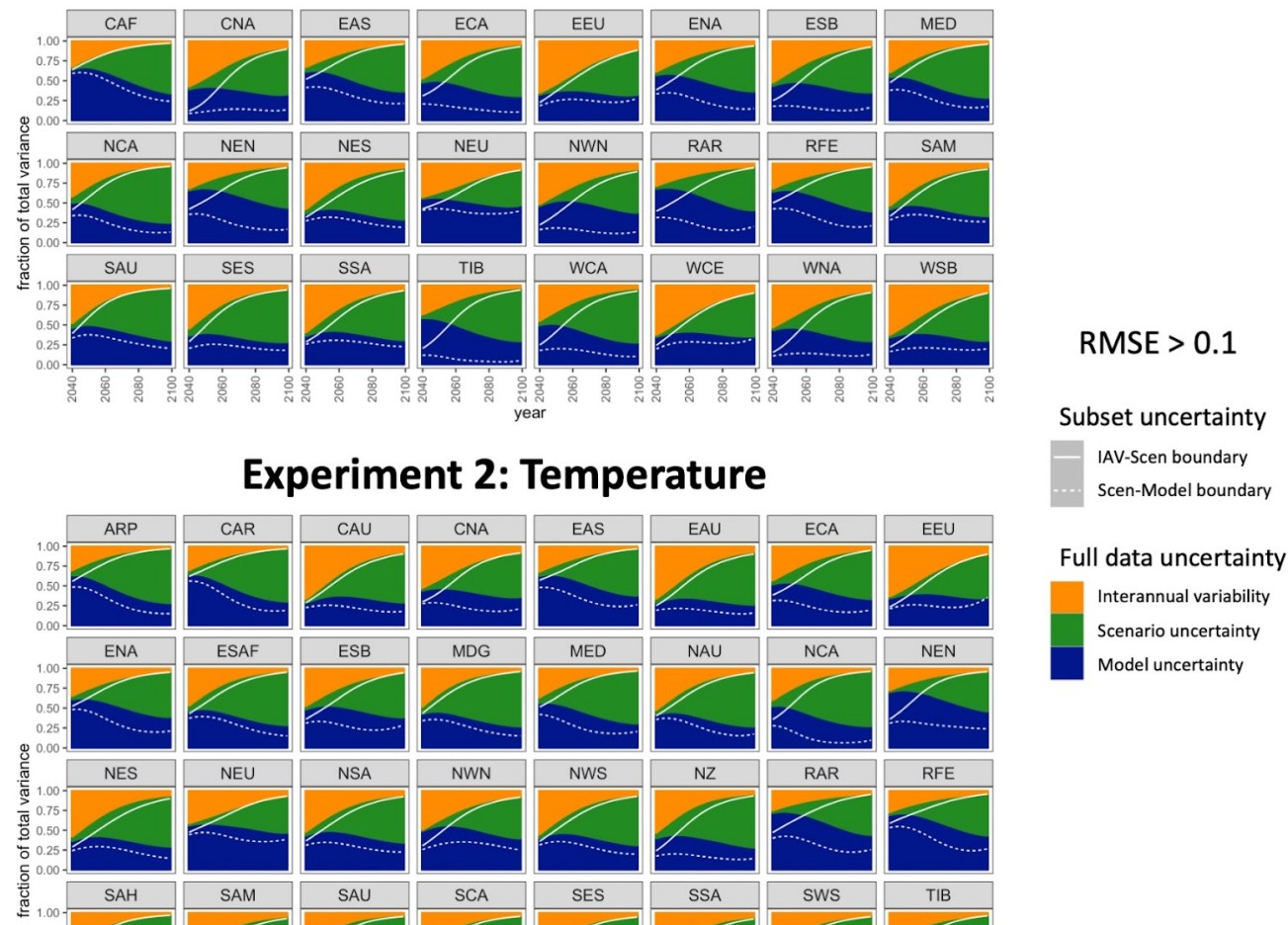

**Figure 6: Regions flagged for closer inspection of their HS fraction time series for temperature. The color-blocked time series are the HS fractions from the full set of data, and the white curves overlaid are the respective boundaries for the subset data's uncertainty partitions.**

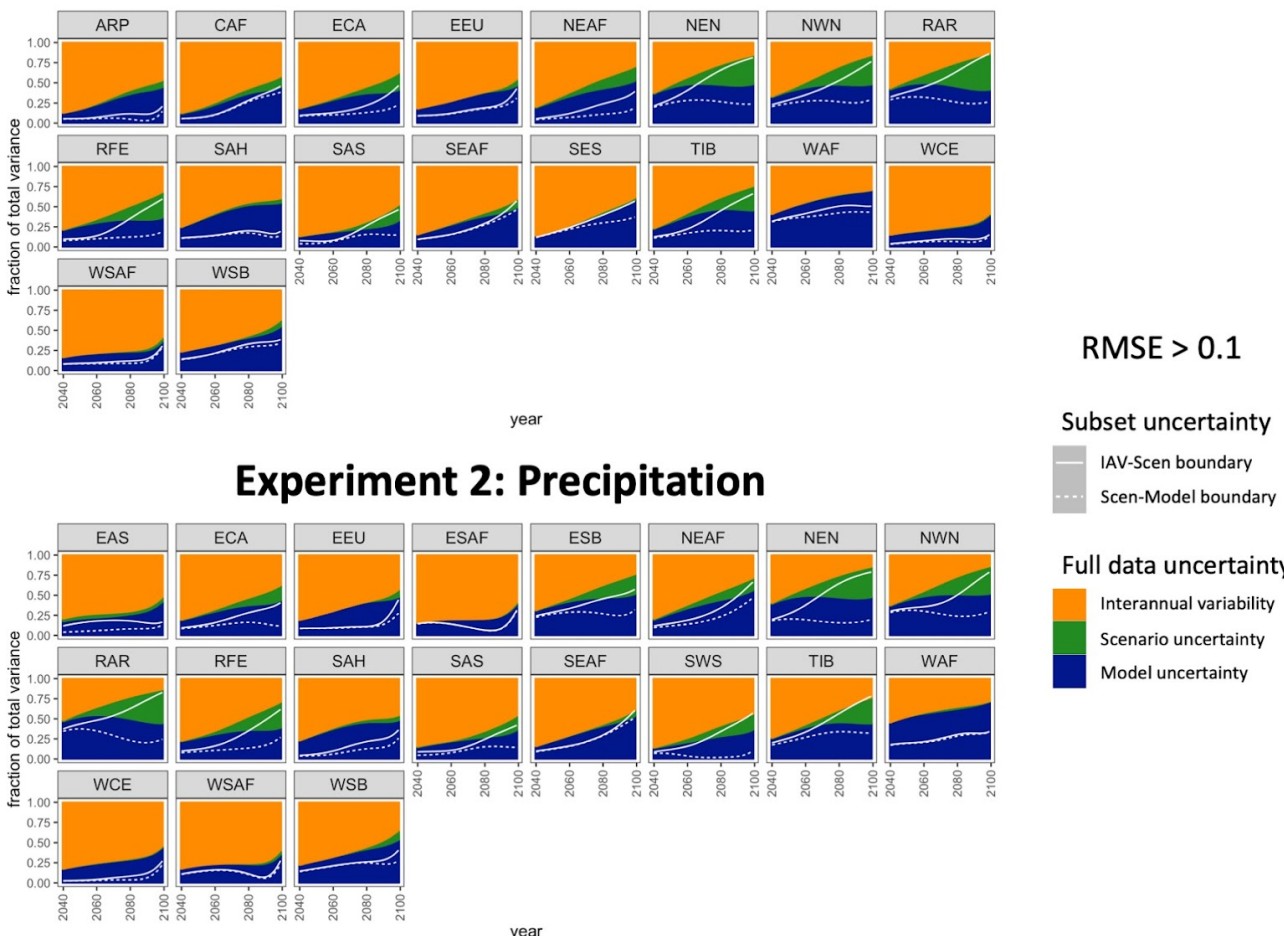

**Figure 7: Same as Figure 6 but for precipitation.**

## 4 Conclusions

This work outlines and documents the success of a method for selecting a subset of climate models from CMIP6 that overall preserve the uncertainty characteristics of the full CMIP ensemble, particularly for use with multisectoral dynamics models that require global coverage and consistency across regions. The methodology is not focused on advocating for a particular set of models as superior, instead focusing on managing uncertainty. Our methodology relies on pre-identifying regional indices of behavior for ESM/GCM output variables, as well as other filters (such as preserving the IPCC

distribution of ECS values) judged to be critical for the robustness of impact and multisectoral
modeling. With these assumptions, far fewer climate inputs are needed to span the range of
uncertainties seen in CMIP6, resulting in fewer impact model runs needing to be performed and
analyzed. There are likely many situations in which a modeler could adapt the details of the method
(outlined in Table 1) and code for their purposes, re-run to identify a subset of climate models, and
validate that new subset in much less time and with much fewer computing resources needed than
simply running impact models with all scenarios and ensemble members available for the 22
ESM/GCMs documented in Table 2. For multisectoral modelers integrating multiple different impacts,
or running large ensemble experiments, the time saved only grows, even when accounting for method
adjustment and re-validation of results.  For researchers focused on emulators, there may be
opportunities to identify fewer climate models that would benefit from generating more initial condition
ensemble members, focusing efforts.  Finally, Earth system modelers can gain new insights into their
individual climate models by adding the approach to uncertainty characterization outlined in this work
to their existing analyses.

The methodology outlined in this paper is an adaptable approach to both retain the major uncertainty
characteristics of a large collection of global-coverage climate model data and to make changes (as we
did to the full ensemble ECS distribution). While there are resulting regions for both temperature and
precipitation where the uncertainty partitions of the subset of ESM/GCMs differ from the full set of
CMIP6 models, these differences are primarily expected based on the different ECS distribution
represented by our subset ESM/GCMs compared to the full data.  For those interested in using our
chosen subset, we hope that by providing detailed information about where the subset differs in Figures
5-7, impact modelers may be able to infer how results would change if the full set of data were used,
with far lower computational burden than running all available data. Further, because the method is
adaptable, an impact modeler particularly interested in a specific region could weight the outcomes in
that region more heavily for selection of the subset.
As noted, this work is primarily coming from the perspective of a multisectoral dynamics modeler
requiring global coverage of a range of climate model output variables at different time scales, and
naturally other perspectives will come with their own caveats. Impacts can be estimated and worked
with at a range of spatial scales; impact modelers concerned with finer scale or local impacts, or
modelers focused on a single region rather than global coverage, may very well be served by
prioritizing other factors like skill in their climate model subselection. Bias correction and downscaling
are also tools heavily used to get to these finer spatial scales, and these processes introduce their own
sources of uncertainty, particularly for very local phenomenon and over complex terrain (Kendon et al.
2010; Mearns et al. 2013; Barsugli et al. 2013; Lafferty and Sriver 2023). Generally, the method
outlined in this work is more appropriate to work with raw CMIP6 data in its native resolutions or an
ensemble of bias-adjusted and downscaled climate data that has been processed using a consistent bias-
adjustment and downscaling method. On a final note for adaptations of this method, we focused on
temperature and precipitation because many variables used in impacts modeling are correlated to or
derived from these variables. This is especially true in agriculture, e.g. Sinha et al. 2023; Sinha et al.
2023; Peterson and Abatzoglou 2014; Allstadt et al. 2015; Gerst et al. 2020, although it holds in other

sectors as well. One area for potential expansion of this method that would have more direct relevance to those derived variables would be to incorporate a time dimension more explicitly.

## Appendix A Additional figures

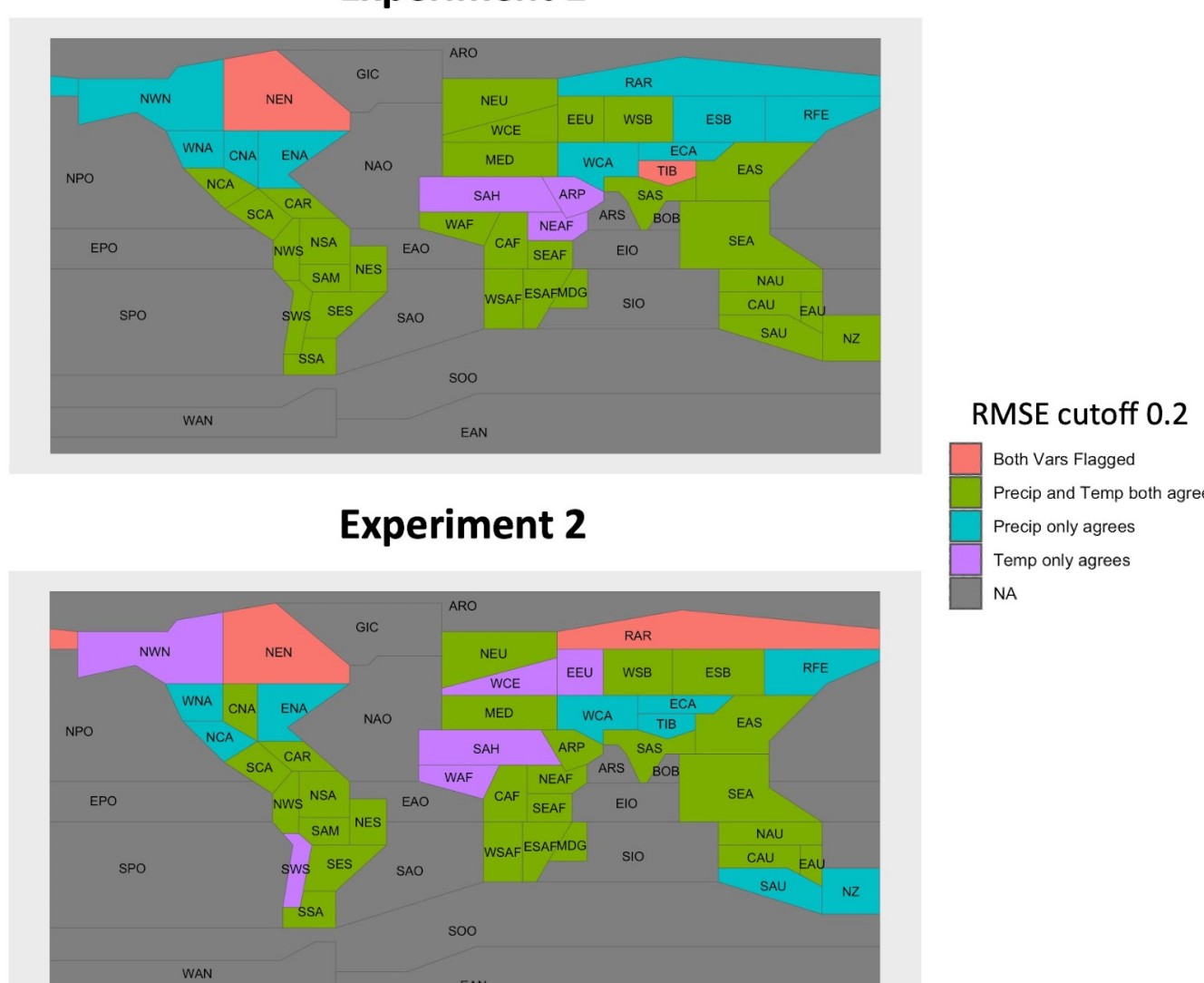


**Figure A1: A color-coded map of regions where temperature, precipitation, both, or neither have RMSE <= 0.2 for all three uncertainty partitions.**

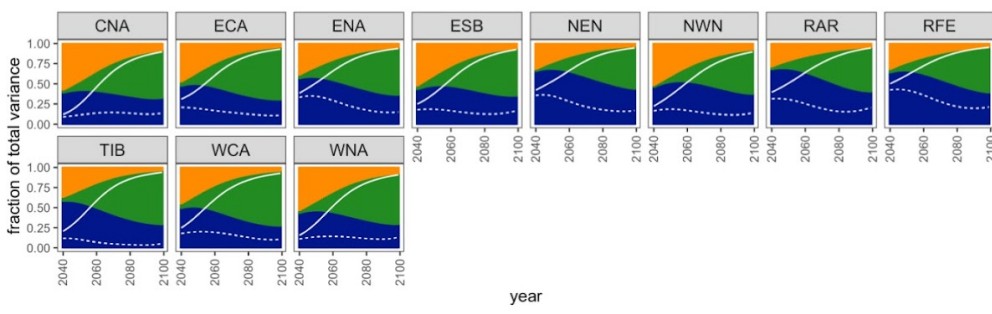


**Figure A2: Same as Figure 6 but for RMS > 0.2 rather than 0.1**

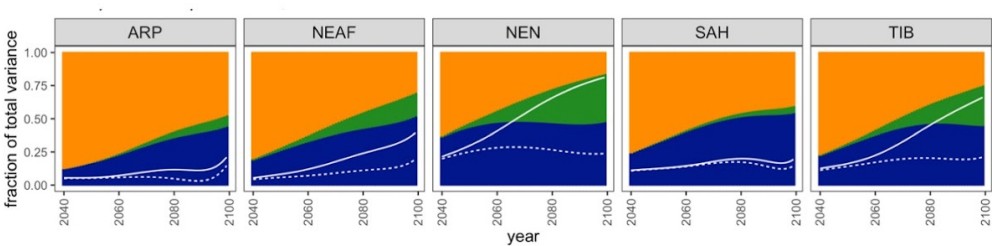

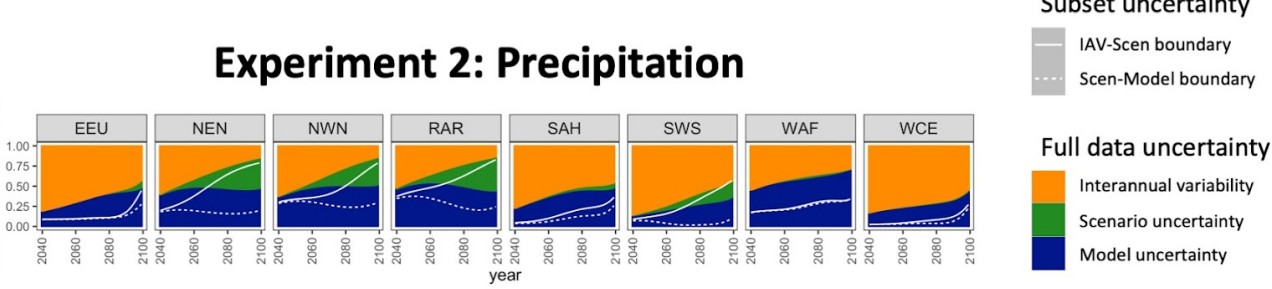

**Figure A3: Same as Figure 7 but for RMS > 0.1 rather than 0.1.**

**Code and data availability:** All code and data are available via a Github metarepository (https://github.com/JGCRI/SnyderEtAl2023_uncertainty_informed_curation_metarepo) and minted with a permanent DOI (https://doi.org/10.57931/2223040)

**Author contributions:** CT conceived of the project, AS led design of the methodology and performed analysis, NP performed analysis, KD provided data; all authors contributed to methodology, analysis, and the writing of the paper.

**Competing interests:** The authors declare that they have no conflict of interest.

**Acknowledgements:** This research was supported by the U.S. Department of Energy, Office of Science, as part of research in MultiSector Dynamics, Earth and Environmental System Modeling Program. The Pacific Northwest National Laboratory is operated for DOE by Battelle Memorial Institute under contract DE-AC05-76RL01830. The views and opinions expressed in this paper are those of the authors alone.

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
