# Peer review of "Uncertainty-informed selection of CMIP6 Earth System Model subsets for use in multisectoral and impact models"

_Earth System Dynamics, 2023_

## Author Comment (AC1)

Uncertainty-informed selection of CMIP6 Earth System Model subsets for use in multisectoral and impact models - Response to Reviewers

**Reviewer 1**

The study posits a strategy for selecting 5 CMIP6 GCMs that are suitable globally for impact model applications based on temperature and precipitation characteristics and the IPCC likely ECS range.

Authors' response [blue throughout]: Thank you very much for your thoughtful review.

The results would benefit from context, both in terms of how the study compares to previous model subselection exercises (why select the same set of models for all regions?)

We have extended our motivation of the work in the Introduction and Methods Sections, noting that: "Specifically, this work is intended for use with models like GCAM, which require global coverage."

Mixing and matching ESMs across regions would result in inconsistent scenarios in that context. We have also added the clarification that if one were interested in specific regions, they could consider data from only those regions.

and in a deeper dive into the origins of the IPCC likely ECS range (where it comes from, what constraint assumptions are being made).

We have added additional discussion of the IPCC ECS range to our Methods section, as this is a critical component of the selection process.

Additionally, the methodology is hard to follow in the appendix and is worth moving to the main text.

We have moved the description of the Hawkins and Sutton values to Methods, and clarified that it is only used for independent evaluation of the model subset selection, rather than as part of the selection process. We have also clarified details of the HS calculations, as raised in points below and by other reviewers.

The primary issue I have, though, is that taking the "model uncertainty, scenario uncertainty, and interannual variability of the full CMIP6 ESM results" is inappropriate in an ensemble of opportunity like CMIP6 without a careful audit of model dependence.

This is an excellent point. This method is very flexible to different assumptions and we have added an example of incorporating considerations about model dependence, and

we have adjusted the Introduction to be clearer that this paper is describing a method for subselection rather than a Universal Best Subset.

Recommended Literature:

Abramowitz, G., Herger, N., Gutmann, E., Hammerling, D., Knutti, R., Leduc, M., Lorenz, R., Pincus, R., and Schmidt, G. A.: ESD Reviews: Model dependence in multi-model climate ensembles: weighting, sub-selection and out-of-sample testing, Earth Syst. Dynam., 10, 91–105, https://doi.org/10.5194/esd-10-91-2019, 2019.

Brands, S.: A circulation-based performance atlas of the CMIP5 and 6 models for regional climate studies in the Northern Hemisphere mid-to-high latitudes, Geosci. Model Dev., 15, 1375–1411, https://doi.org/10.5194/gmd-15-1375-2022, 2022.

Merrifield, A. L., Brunner, L., Lorenz, R., Humphrey, V., and Knutti, R.: Climate model Selection by Independence, Performance, and Spread (ClimSIPS v1.0.1) for regional applications, Geosci. Model Dev., 16, 4715–4747, https://doi.org/10.5194/gmd-16-4715-2023, 2023.

Thank you, we have incorporated these citations into our Introduction to further highlight that the method we propose can serve as a complement to other selection criteria.

Specific Comments:

L60-62: "In a world unburdened by time and computing constraints, an impact model would take as input every projected data set available to have a full understanding of possible outcomes." - An ensemble of every projected data set in CMIP6 does not confer the full understanding of possible outcomes. It would include 50 initial condition ensemble members of certain ESMs and one ensemble member for others. Is the first 50x more likely to be true? Beyond the unequal voting power of the large ensembles in CMIP6, the ensemble contains a number of "hidden dependencies": models with different names but near-identical code. For uncertainty to mean "our full understanding of possible outcomes", model dependence must be handled properly.

As noted above, we have added a discussion of addressing model dependency via an additional example. To address the over-weighting of models with more ensemble members, we have also clarified the region-level indices used to characterize each model's outputs in the Methods. Briefly, the indices are calculated on each ensemble member available for a given model, but then the ensemble average of the index is the only value used for a model. In other words, models with 50 ensemble members don't get 50 entries for each index in each region. They do, however, likely have a more

**confident estimate of the ensemble averaged index than say a model with only 1 ensemble member (particularly for internal variability).**

L81-82: While this is an interesting objective, the size of the initial condition ensembles submitted to an exercise like CMIP is a function of computational resources and goodwill (they are submitting "free" data for others) on the side of the modeling centers. It is beneficial to many researchers that CMIP is inclusive and does not "pick favorites" thus encouraging participation.

This is a great point and it was not our intention to suggest favorites should be picked. We have adjusted this sentence. We have also removed our suggestions to CMIP modelers to leverage this idea to more efficiently deploy total computational resources. While this is technically possible to take from our work, it's also not very realistic to the CMIP process for the reasons you highlight.

Table 1: Of the 22 models you are using, 6 are connected, either by legacy or because they use a version, to NCAR's Community Atmosphere Model (CAM) development cycle. As this leaves the potential for CAM to have more influence on your uncertainty benchmark, the choice must be discussed. Additional similar models, such as ACCESS-CM2 / UKESM1-0-LL and MPI-ESM1-2- HR / MPI-ESM1-2- LR, present could be creating a "rather heterogeneous, clustered distribution, with families of closely related models lying close together but with significant voids in-between model clusters" (description of CMIP5 from Sanderson, B. M., Knutti, R., and Caldwell, P.: A representative democracy to reduce interdependency in a multimodel ensemble, J. Climate, 28, 5171–5194, https://doi.org/10.1175/JCLI-D-14-00362.1, 2015.). Could multiple similar models elevate outliers (e.g., MIROC) in your metric?

The metric is actually intended to capture outliers, although MIROC is not an outlier in this group. If we only had one representative model from each dependent group, it's possible the eigenvectors would change enough that MIROC would become an outlier in that projection space and therefore the metric would select in that case.

L166: Please justify the citation of Scafetta, 2022. See <a href="https://aqupubs.onlinelibrary.wiley.com/doi/10.1029/2022GL102530">https://aqupubs.onlinelibrary.wiley.com/doi/10.1029/2022GL102530</a>

Thank you for bringing this to our attention. We only used this to source ECS values for two models (CMCC-CM2-SR5 and NorESM2-MM). We have removed these values and references to Scafetta from the manuscript. Like other models we could not track down ECS values for (e.g. EC-Earth3-Veg-LR), these models are included as part of the full space of model behaviors we characterize with PCA, but they are not eligible for selection to the subset.

In step 3 of Table 2, how are you computing an ensemble average for the models that only provide a single run?

The ensemble average is the average of that one realization, ie that realization's value. We have clarified this in the text.

L224: Omit "over time?"

**Thank you, done.**

L315: "Models who more closely match the trend of observational data (W5E5v2.0 (Lange et al., 2021)) over the historic period will have their observations hold more weight. " Why? Trends are highly sensitive to internal variability, which is inherently random in temporal phase, i.e. no reason a model and observation should have the same sequence of it. A match in trend between observations and a model over a particular time period often occurs by chance and is not indicative model performance.

Deser, C., Phillips, A., Alexander, M. A., and Smoliak, B. V.: Projecting North American climate over the next 50 years: Uncertainty due to internal variability, J. Climate, 27, 2271–2296, https://doi.org/10.1175/JCLI-D-13-00451.1, 2014.

This is an excellent point. We initially used the weighting according to matching observational data only in the Hawkins and Sutton calculations we perform to evaluate the success of our subset, not in actually selecting the subset itself. We did so following the inclusion of similar weighting in the original HS papers; the authors however do note that they did not expect uniform weights to dramatically change the breakdown of total variance. Also, we do not see this method as relating to model validation and reliability (which hard to characterize for global scale applications of multiple variables), but simply as a way to subset the large ensemble while preserving uncertainty characteristics. In this spirit, we move to uniform weights in all of our HS calculations for evaluation.

**Reviewer 2**

Review of "Uncertainty-informed selection of CMIP6 Earth System Model subsets for use in multisectoral and impact models" by Snyder et al.

The presented study develops a model selection procedure that aims to preserve the distribution of total uncertainty as defined by Hawkins and Sutton. The motivation and need for some kind sub-selection of global climate models for impact models is clear and well-motivated by the authors. Their method is fairly straight-forward and is shown to work for the metric the authors use as validation.

I think this approach could be valuable contribution to be used as an objective selection criterion if the authors manage to manage to better describe the caveats and limits of this approach and better showcase the effect of the sub-selection. In its current state I can not recommend the manuscript for publication based on the major comments outlined in the following.

Thank you, we respond to each comment below, in blue.

**Major comments**

I am very critical about the idea of optimizing a sub-selection method only for the distribution of the three sources of uncertainty. I note that the authors also manually select by ECS but this seems to be not really part of the method and is presented more as an afterthought.

Thank you, we have edited the text to more heavily emphasize the role of ECS in the Methods, which was always a critical aspect of our approach but evidently was not highlighted sufficiently in its importance.

In any case, such an approach opens up the risk of selecting highly dependent models (as seems in fact to be the case in the presented example with both ACCESS versions being part of the suggested sub-ensemble) as well as objectively bad models. Both are situations which should be avoided I would argue. In fact, in a recent study Merrifield et al. (2023; 10.5194/gmd-16-4715-2023) set out to solve a similar problem but also consider model performance and independence. Their results should at least be compared to the results presented in this manuscript.

Thank you - we have added a citation to Merrifield et al as well as a second experiment to the manuscript to highlight the flexibility of the method. In the second experiment, we consider an extremely simple definition of independent models to focus on, and repeat the selection process. As you are aware, there are a wide variety of definitions and criteria to define model independence, and the question is not yet settled in the literature. We hope that while our definition of independence in our added Experiment 2 is simple, it highlights that the method *could* be applied and validated for a different set of independent models making up the 'full data' based on a different definition of independence. In other words, we think our method can be a powerful complement to other methods concerned more heavily with independence. At the very least, a researcher performing this experiment would likely learn something about their definition of independence.

I have to admit I do not fully understand the role of the scenario uncertainty this manuscript. In an idealized case the scenario uncertainty as defined by HS is only dependent on the scenarios used. Why is it considered here, what is it relation to model selection? If it is sensitive to the selected (ECS preserving) model ensemble this is only a sign that the HS method is not properly able to isolate forced response, model uncertainty, and internal variability (as discussed in Lehner et al. 2020) is it not?

Thank you for highlighting this. It is true that we are looking to represent model uncertainty which is reflected in both T&P anomalies and IASD. But we also wish to do so in a robust way, and for the CMIP6 ensemble that means we need to consider the scenarios that were run (at least the Tier1 scenarios, here). We aren't sub-selecting for scenarios but the scenario space is vital to appreciate the variability of the different models. We have attempted to clarify this in the methods section describing the HS fractions and what we use them for.

With respect to the verification metric, I am not sure if optimizing for the relative distribution of uncertainty is really ideal. At least the authors should consider also looking at the absolute position of their selected models in temperature-precipitation change space. I could imagine a situation where the relative distribution of uncertainties in the subset is very similar to the full ensemble without the subset being representative when considering, e.g., absolute changes.

We have moved the description of the Hawkins and Sutton values to Methods, and clarified that it is only used for independent evaluation of the model subset selection, rather than as part of the selection process. We have further emphasized that the selection process only takes place in the absolute position/changes space of T and P anomalies/IASD and not the relative space of the HS fractions.

Minor comments

line 33: "Scenario simulations from CMIP (most recently through ScenarioMIP, (O'Neill et al. 2016)"

closing braked missing

Thank you, fixed.

36: "Using such multi-model ensembles captures the process and structural uncertainties represented by sampling across ESMs, scenario uncertainty,"

Minor point but a multi model is not necessary to represent scenario uncertainty?

Similar statement also in the abstract: "In this work, we present a method to select a subset of the latest phase, CMIP6, models for use as inputs to a sectoral impact or multisectoral models, while still representing the range of model uncertainty, scenario uncertainty, and interannual variability of the full CMIP6 ESM results"

Thank you for highlighting this. It is true that we are looking to represent model uncertainty which is reflected in both T&P anomalies and IASD. But we also wish to do so in a robust way, and for the CMIP6 ensemble that means we need to consider the scenarios that were run (at least the Tier1 scenarios, here). We aren't sub-selecting for scenarios but the scenario space is vital to appreciate the variability of the different models. We have attempted to clarify this in the methods section describing the HS fractions and what we use them for.

45: "For Earth system modelers, variability across ESMs' projections of future climate variables can be significant (Hawkins and Sutton 2009; Hawkins and Sutton 2011; Lehner et al. 2020) and so the participation of multiple modeling centers running multiple scenarios is critical to understanding the future of the Earth system."

I agree with this statement but "variability across ESMs" is a bit vague. Of the three sources of variability considered in the cited studies only one really needs a multi-model ensemble. I am also not sure if these are really the right studies to cite here as they mainly look at relative contributions of individual sources of uncertainty.

We don't entirely understand what the reviewer means with this comment. However, we do not think it is possible to robustly assess uncertainty in internal variability or scenario without a multi-model ensemble. We are happy to respond to and address any clarifications on this comment.

57: "However the total burden across modeling centers to sample across ESMs and scenarios still remains high, even with this potential efficiency." I am not sure I understand the first part of this sentence?

Even if emulators can lighten the computational effort in scenario space, modeling centers are still addressing competing demands from different experiments' goals. We have attempted to clarify in the introduction.

72: "these competing priorities" what are the 'competing priorities' here?

We have attempted to clarify this text.

Tab 1: I am a bit confused about the use of the word ESM in this manuscript. This table clearly also contains models versions with are not ESMs but GCMs? (ACCESS-CM2 for example)

Thank you for pointing this out, we have corrected the language throughout the manuscript to refer to the collection of models simply as 'the models' when a shorthand is desired, and we have clarified that both ESMs and GCMs are considered. We have also reminded readers that even Earth System Models are run in concentration-driven mode rather than emissions-driven mode for these experiments in CMIP6.

108: "and the interannual standard deviation"

It is unclear to me how this is done or in which time-period.

We have clarified this in the text: "Interannual standard deviation is calculated by detrending the regional average level temperature and precipitation time series from 1994-2100 using non-parametric locally weighted smoothing (LOESS as implemented in the python statsmodels package), and then taking the standard deviation of the residuals.

114: At this point I am also wondering how initial-condition members are considered exactly? Is seems that the indices would be sensible to the number of ensemble members?

We have clarified our description of the indices in the text: "For each scenario, region, and available ensemble member in each climate model we extract the following

temperature and precipitation outputs: mid-century (2040-2059) average anomaly relative to that model's historical average (1995-2014), the end of century (2080-2099) anomaly relative to historical average, and the interannual standard deviation. Interannual standard deviation is calculated by detrending the regional average level temperature and precipitation time series from 1994-2100 using non-parametric locally weighted smoothing (LOESS as implemented in the python statsmodels package), and then taking the standard deviation of the residuals. For each scenario and model, these ensemble-member values are used to calculate the ensemble average to form our final indices in each region."

Please also see our response to R1 in regards to a similar quest: "Briefly, the indices are calculated on each ensemble member available for a given model, but then the ensemble average of the index is the only value used for a model. In other words, models with 50 ensemble members don't get 50 entries for each index in each region. They do, however, likely have a more confident estimate of the ensemble averaged index than say a model with only 1 ensemble member (particularly for internal variability)."

Fig 1: this figure seems unproportionally big for the amount of information it conveys.

Thank you for this suggestion, we have decreased the size to accommodate the scree plots for both experiments we now discuss in the text, and (briefly) further explained why we check the scree plots to determine how many eigenvectors to retain.

Fig 3: I am not quite sure what information this figure is supposed to convey to the reader? In particular, since it is hardly discussed in the text. For example, is it somehow relevant that MPI or MRI (hard to distinguish) seem to behave quite different in PC2?

Thank you - we have added more motivating text to where this figure is presented and throughout the methods.

165: "Core Writing Team & (eds.)," strange citation

We have adjusted the citation in the bib file so the editors' names are included but the IPCC requested citation is indeed a little odd: "IPCC, 2023: Summary for Policymakers.

In: Climate Change 2023: Synthesis Report. Contribution of Working Groups I, II and III to the Sixth Assessment Report of the Intergovernmental Panel on Climate Change [Core Writing Team, H. Lee and J. Romero (eds.)]. IPCC, Geneva, Switzerland, pp. 1-34, doi: 10.59327/IPCC/AR6-9789291691647.001"

How many subsets fulfill this criterion?

We have added this number - out of 22 choose 5 potential subsets, there are 72 subsets available meeting this constraint using all models for which we could identify reliable ECS values.

Fig 4: Again I am not sure what I learn from this plot.

Thank you, we have expanded our discussion of this plot.

199: "To manage the inspection of three time series" Is 'three time-series' referring to Hawkins and Suttons model, forced, and internal components? This is not clear to me here. If yesI would note that in HS internal variability is not a timeseries but constant over time if memory serves.

We have clarified that the three time series is referring to the fraction of total uncertainty described by the model, forced, and internal components. The HS internal variability is itself a single value, but that single value makes up a different fraction of the total uncertainty as the model and forced components vary over time. We have clarified in the text.

223: "For temperature, we see that interannual variability is often performing well, with increasingly better performance over time? over time." Typo

Thank you, corrected.

**Reviewer 3**

Review of esd-2023-41

Title: Uncertainty-informed selection of CMIP6 Earth System Model subsets for use in multisectoral and impact models

Authors: Abigail Snyder, Noah Prime, Claudia Tebaldi, Kalyn Dorheim

**Overall Recommendation: Minor Revisions**

This study, submitted to Earth System Dynamics, highlights a new approach to select subsets of ESMs (and potentially GCMs) for use in multi-sectoral and impacts modeling. The approach is novel and attempts to capture multiple sources of uncertainty in the subset and does have potential utility for multiple applications beyond what was mentioned in the manuscript. However, I do think the manuscript could be improved by added justification in a few places and addressing some particular aspects from prior literature that is relevant. In particular, the attempt at subset selection itself is not a new effort and the authors should place this manuscript in the context of prior attempts at subset selection. My general comments to improve the manuscript are below, followed by specific comments. I look forward to seeing the revised manuscript in print.

Thank you, responses in blue to each comment below.

**General Comments**

There are a number of current issues in literature related to the selection of ESMs (or even just a set of climate projections) for use in applications that were not discussed here but are quite relevant.

First and foremost, the challenge of ensemble subset selection is not new. It would be worthwhile given the focus on using ESMs in impacts models to place this article in context with existing literature. For example, Parding et al (2020) produced the GCMeval tool, which is designed as an interactive tool for evaluation and selection of climate model ensembles for use in multiple applications. The methods in the GCMeval tool are relatively simplistic. In the context of this manuscript, I am left to wonder how different this approach is to the approach of Parding et al (2020) or other previous research. It would strengthen the manuscript to briefly discuss the differences between this approach and others in prior literature. Ideally, there should be some analytic

comparisons between these approaches, but I believe a brief discussion of the other literature on this topic would suffice for this manuscript. In addition, the authors are addressing the "practitioner's dilemma" (Barsugli et al. 2013). Though it was not mentioned by name, it may also be worthwhile to discuss how this approach builds on the literature associated with this well-known challenge.

Thank you for these recommendations - we have added further discussion of these in our introduction and throughout the manuscript. We have clarified the framing of our manuscript as being an uncertainty-first focused method that can be used potentially in complement to methods for evaluation and model dependence such as those in Parding et al, Merrifield et al, and others. We have also clarified in our framing that the primary use-case in mind for this method is for models that require global coverage of climate data (such as GCAM and similar models) rather than more local phenomenon. Discussion of downscaling and the practitioner's dilemma has been expanded in our conclusions.

Second, the approach in this manuscript may well retain "hot-models", particularly as one goes to large subset sizes. The literature regarding if one should use hot-models is mixed, ranging from omitting hot models entirely (Hausfather et al. 2022), to down-weighting hot models based on ECS (Massoud et al. 2023), or simply keeping hot models as they may not have serious impacts on impacts modeling (Asenjan et al. 2023). It would be worthwhile for the authors to address how their method handles the "hot models" problem specifically, when their approach may indeed retain these models in their subsets.

Thank you for this note. In response to this and comments from the other reviewers, we have more strongly emphasized the role of constraining our model subset to match the IPCC most likely distribution of ECS values. This of course allows one high ECS model to be selected in the subset to explore potential 'worst case' impacts downstream, but overall moves the 'hotness' of our subset to be lower than the full collection of voluntarily submitted models. In addition to emphasizing this more clearly in the methods, we incorporate this into the discussion of our results as well.

Third, the focus is on impacts modeling with ESMs, but this is not the only use for ESM output. Depending on the needs of a stakeholder, one may not need additional modeling, but rather require complex variables derived from ESM output. A prime example of this is the use of spring phenological indices to determine projected changes in first leaf, first bloom, or the likelihood of a false spring (Gerst et al. 2020; Allstadt et al;

2015; Peterson and Abatzoglou, 2014). The approach in this manuscript could well be used in situations that don't require additional modeling per se, but do require derivation from ESM or downscaled ESM output. Such literature should be briefly mentioned with the authors comment.

**Thank you, we have added a discussion of considering more derived variables to our conclusions.**

Fourth, the Hawkins and Sutton (2009) approach does have some valid criticisms. I suggest looking at the recent work of Lafferty and Sriver (2023), which also uses Hawkins and Sutton (2009) and the work of Wootten et al. (2017). The Lafferty and Sriver (2023) article addresses the critiques around the Hawkins and Sutton (2009) approach.

While we agree that no method is perfect, the Hawkins and Sutton approach remains a useful framework for the qualitative evaluation of how well the selected subsets of climate models preserve uncertainty characteristics of the full data. We find it especially useful for understanding whether discrepancies in breakdown of total variance for the full vs subset data are explainable by our choice to constrain ECS distribution as part of the selection process as a way to address the hot model problem. We have adjusted the text to emphasize these points, and that the Hawkins and Sutton breakdown are not an active part of the selection procedure but rather a post-hoc evaluation tool, throughout the text. We have also clarified in the introduction and conclusions that this method is meant more for studies using multisector dynamic models that require global coverage, where consistency in handling regions is key. We have also added citation to Lafferty and Sriver in our discussion of how this work interacts with bias correction and downscaling considerations in the conclusion, in response to this and the next comment.

Finally, while the selection of ESMs is important, most impacts modelers do not use the ESMs directly, but use the downscaled ESM output (whether dynamically or statistically downscaled). The authors mention this briefly in passing, but it is important to acknowledge this in the conclusions also. Downscaling is itself another source of uncertainty, so it is a question of if this approach could also be applied with ESMs downscaled with multiple approaches.

We have added a brief paragraph to the conclusions regarding bias correction in downscaling. In general, we do not recommend applying this method be applied to already-downscaled data that has been downscaled by different methods. While bias-correction and downscaling is expensive, we believe the consistency gained from using a single robust method for all climate data sets is worth this. Indeed, one

motivation for us as users of this method is to only have to bias-correct and downscale the output of 5 models instead of 22 using the ISIMIP3b protocol.

Literature mentioned above:

Allstadt, A. J., S. J. Vavrus, P. J. Heglund, A. M. Pidgeon, W. E. Thogmartin, and V. C. Radeloff, 2015: Spring plant phenology and false springs in the conterminous US during the 21st century. *Environmental Research Letters*, 10, https://doi.org/10.1088/1748-9326/10/10/104008

Asenjan, M.R., F. Brissette, J.-L. Martel, and R. Arsenault, 2023: Understanding the influence of "hot" models in climate impact studies: a hydrological perspective. *Hydrology and Earth System Sciences*, 27, 4355-4367, DOI: 10.5194/hess-27-4355-2023

Barsugli, J., and Coauthors, 2013: The Practitioner's Dilemma: How to Assess the Credibility of Downscaled Climate Projections. *Eos Transactions*, 94, 424–425, https://doi.org/10.1002/2013EO460005.

Gerst, K. L., T. M. Crimmins, E. E. Posthumus, A. H. Rosemartin, and M. D. Schwartz, 2020: How well do the spring indices predict phenological activity across plant species? *Int J Biometeorol*, 64, 889–901, https://doi.org/10.1007/s00484-020-01879-z.

Hausfather, Z., K. Marvel, G. A. Schmidt, J. W. Nielsen-Gammon, and M. Zelinka, 2022: Climate simulations: recognize the 'hot model' problem. *Nature*, 605, 26–29, https://doi.org/10.1038/d41586-022-01192-2.

Lafferty, D.C. and R.L. Sriver, 2023: Downscaling and bias-correction contribute considerable uncertainty to local climate projections in CMIP6. *Nature Partner Journal – Climate and Atmospheric Science*, 6, doi: https://doi.org/10.1038/s41612-023-00486-0

Parding, K. M., and Coauthors, 2020: GCMeval – An interactive tool for evaluation and selection of climate model ensembles. *Climate Services*, 18, 100167, https://doi.org/10.1016/j.cliser.2020.100167.

Peterson, A. G., and J. T. Abatzoglou, 2014: Observed changes in false springs over the contiguous United States. *Geophysical Research Letters*, 41, 3307–3314, https://doi.org/10.1002/2014GL061184.Received.

Wootten, A., A. Terando, B. J. Reich, R. P. Boyles, and F. Semazzi, 2017: Characterizing Sources of Uncertainty from Global Climate Models and Downscaling Techniques. *Journal of Applied Meteorology and Climatology*, 56, 3245–3262, https://doi.org/10.1175/JAMC-D-17-0087.1.

Minor Comments:

Line 84, first mention of Table 2: Table 2 defines the process for selection to aid the reader, yet it is positioned in the text far from the section. It's also mentioned before Table 1 and Table 1 appears sooner in the text. I suggest reordering Table 2 and Table 1 and placing the renamed Table 2 earlier in the text to help the reader.

Thank you, we have done this.

Line 105: "For each scenario and region in each ESM,..." – Am I correct that the averages calculated are across all the initializations of each ESM? Or is it the average across all models?

Yes, the average is across all initializations of each model within each scenarios, not across models. We have clarified this in the revised methods.

Lines 127-128: "Based on this figure,...explaining 71.8% of variance." – Why only 5 eigenvectors? Why not more?

Thank you, we have added text explaining that we cut eigenvectors just after the major 'elbow' in the scree plot, which is a pretty common rule of thumb albeit there are no hard and fast rules. There's no reason this method couldn't be applied on more, or all, eigenvectors. But for the sake of faster evaluation and easier visualization, we stop at 5.

Line 224: "...better performance over t ime? over time." – This is a typo.

Thank you, we have corrected this.

Lines 220-240: It seems the discussion focused on temperature plots only, but I didn't see any comparison of temperature vs. precipitation results.

Thank you, we have added some discussion of this.

---

## Referee Report (RR1)

**General Remarks**

This study has benefitted from the first round of revisions, and I find the methodology much easier to follow as part of the main text. It is an interesting approach to model selection; the uncertainty partitioning aspect clearly took a lot of effort and will be useful for many end-users of CMIP6.  However, I am searching for more assistance in interpreting the results from Figures 2 through 7 in the text. I'm not sure what aspects of the figures I should be looking at, there is very little discussion of the similarities and differences between Experiment 1 and Experiment 2, and I am missing the justification for why the selected subset is superior to other possible subsets. It may be too much to compute Figure 5 for all 72 subsets you are considering, but it would support the selection you've done if there was some comparison between the subset you selected and the ones you did not. I have detailed places where results can be elaborated on further, as well as a few figure style suggestions, in the specific comments portion of this review.

**Specific Comments**

L34: (CMIP; Eyring et al 2016) > in LaTeX, (CMIP; \citealp{Eyring})

L35-36: Same as L34

L38-39: \citep[e.g.][]{X,Y,Z}

L63-64: Same as L38-39

L73-74: Same as L38-39

L135: Table1?

Table 2: ECS values are available for your missing models:

- CMCC-CM2-SR5: Values reported in the IPCC's Assessment Report 6 Working Group I Chapter 7 Supplementary Material (The Earth's energy budget, climate feedbacks, and climate sensitivity) Table 7.SM.5.

- EC-Earth3-Veg-LR and FGOALS-g3: https://github.com/mzelinka/cmip56_forcing_feedback_ecs

- NorESM2-MM: Seland, Ø., Bentsen, M., Graff, L., Olivié, D., Toniazzo, T., Gjermundsen, A., Debernard, J., Gupta, A., He, Y., Kirkevåg, A., Schwinger,  J., Tjiputra, J., Aas, K., Bethke, I., Fan, Y., Griesfeller, J., Grini, A., Guo, C., Ilicak, M., and Michael, S.: The Norwegian Earth System Model, NorESM2 – Evaluation of the CMIP6 DECK and historical simulations, https://doi.org/10.5194/gmd-2019-378, 2020a.

L159-161: Is it fair to compare individual realizations to an ensemble average for things like interannual standard deviation? Additionally, how do you handle the fact ensemble

spread in precip. is much larger than ensemble spread in temperature for many regions?

Figure 1: Can the side-by-side panels be on the same y axis scale? Additionally, the figure titles are identical, is this intentional?

L206: Can you give a sense of what the sign of the PC represents? Could you show the full ensemble variance spatially? That might help with the interpretation of the EOF.

L215: Can you elaborate on "strikingly similar"?

L222: What features?

L235: Can you comment on what is happening with MRI is PC2?
Figure 3 and Figure 4: These scatters are not so legible, and I'm not sure what is to be gained from scattering each PC against the others? Would scatters of each PC against total variance illustrate the message?

L258-260: I think I am misunderstanding. Is the idea to have the subset sit at the center of the distribution? Or to cover the spread? I drew a scenario that I think illustrates my confusion about the summary metric.

[Figure]

L305: stray box in the equation.

Figure 5: Why do you think you lose temperature agreement in so many regions in Experiment 2?

Figure 6: My read here is that in all the regions you flag, model uncertainty is always under-represented by the subset (due to the constraint on ECS you impose) and the partition between scenario uncertainty and interannual variability in the subset approaches the full ensembles over time, but scenario uncertainty of the subset is always under that of the full ensemble in 2040. Is this to be expected? Do we see some cases where the subset has more scenario uncertainty than the full ensemble early in the record? Though model uncertainty is shifted with respect to the full ensemble it seems to evolve through time in a similar way in most cases. Isn't that more important than just a RMSE < 0.1?

Figure 7: Again, I see a difference between cases where the subset is shifted down w.r.t. the full ensemble partition (e.g., Experiment 2 ESB) and cases where the partition is fundamentally different in time (e.g., Experiment 2 EAS). Have you investigated why this might be in more detail?

---

## Referee Report (RR2)

Review of esd-2023-41
**Title: Uncertainty-informed selection of CMIP6 Earth System Model subsets for use in multisectoral and impact models**
Authors: Abigail Snyder, Noah Prime, Claudia Tebaldi, Kalyn Dorheim

*Overall Recommendation: Accept*

This study, submitted to Earth System Dynamics, highlights a new approach to select subsets of CMIP6 GCMs for use in multi-sectoral and impacts modeling. The approach is novel and attempts to capture multiple sources of uncertainty in the subset and does have potential utility for multiple applications beyond what was mentioned in the manuscript. This is the second time I have reviewed this manuscript and I'm pleased that the authors addressed my previous comments. As my remaining comments are minor issues, I recommend moving forward with publication. I look forward to seeing the revised manuscript in print.

Minor Comments:

Line 135 – There's a typo where Tables 1, 2, and 3 are all mentioned in this line. I believe Table 2 was what the author's meant to refer too.

Lines 138-142: "Models for which we… (more details in Section 2.2)." – How do you know the distribution of ECS is preserved when those without an ECS value in literature are removed? Presumably, these models have an ECS value, it is simply that the ECS values for thos GCMs have not been assessed by other literature.

Line 189 – Like the comment for Line 135, there's a typo where Tables 1, 2, and 3 are mentioned in the same line.

Lines 199-201: "Based on this figure…flexibility of this method." – The authors did respond to my question with respect to the number of chosen eigenvectors. I suggest including the response here rather than only in the response to the reviewers.

---

## Author Response (AR2)

**Reviewer 1:**
This is my second review of the manuscript "Uncertainty-informed selection of CMIP6 Earth System Model subsets for use in multisectoral and impact models" by A. Snyder and colleagues.

Thank you to the authors for extending the paper at several places and including more context. I think I understand the authors aim with this method better now.

Overall, I only have small technical comments left and think the manuscript can be published in its current form.

Thank you for your feedback. Our responses are in blue and **all line numbers referenced refer to line numbers in the new track changes manuscript.**

Minor comments

Table 2: The authors could have a look at ECS values from Mark Zelinka, which I think includes the missing models: https://github.com/mzelinka/cmip56_forcing_feedback_ecs
Thank you! We have sourced these values, added this citation, and re-run our analyses to include these models as candidates for selection (they were always included in the set of models making up the full data in each experiment). The final subsets for each experiment were unchanged.

Figure 1: I assume the figure title of the right panel should not read 'all ESMs'?
Also, in the rest of the manuscript, the authors have switched to referring to their model collection as ESM/GCM.
Thank you for catching this, we have corrected this typo.

Figure 2: IASD is never introduced as far as I can tell.
Thank you for catching this, we have added the acronym to the text before Figure 2 where we first note that interannual standard deviation is one of our indices (~L178).

Figure 6: IAV is never introduced.
Thank you for catching this, we have added the acronym to the text where we introduce the Hawkins and Sutton calculations (~L347)

**Reviewer 2:**

**General Remarks**

This study has benefitted from the first round of revisions, and I find the methodology much easier to follow as part of the main text. It is an interesting approach to model selection; the uncertainty partitioning aspect clearly took a lot of effort and will be useful for many end-users of CMIP6. However, I am searching for more assistance in interpreting the results from Figures 2 through 7 in the text. I'm not sure what aspects of the figures I should be looking at, there is very little discussion of the similarities and differences between Experiment 1 and Experiment 2, and I am missing the justification for why the selected subset is superior to other possible subsets. It may be too much to compute Figure 5 for all 72 subsets you are considering, but it would support the selection you've done if there was some comparison between the subset you selected and the ones you did not. I have detailed places where results can be elaborated on further, as well as a few figure style suggestions, in the specific comments portion of this review. Thank you for taking the time reviewing the revised manuscript. Our responses are in blue and **all line numbers referenced refer to line numbers in the new track changes manuscript.**

With regard to the above, we have clarified the goal of our selection criteria in the manuscript (L112-122), and the selected subset of models meets that criteria the best compared to other possible subsets: to first represent (via PCA) a space that maximizes total variance of the full set of data in each experiment and to second select models that cover the range of that space while also preserving the IPCC distribution of ECS values. We are not evangelizing about a particular subset of models being the best but we are proposing a method that can be applied to new sets of simulations/models/regional discretizations and showing how it works for the choices outlined in Table 1. We have added text to our conclusions to re-emphasize this at the close of the paper (L473-474). We added the "post-facto" description of how our selection compares to the full ensemble when the Hawkins and Sutton partition is performed as a sanity check, looking for overall agreement across regions. We submit that the overall result is satisfactory (across regions and the two variables). Of course there will always be some specific region where the subset of models does not closely represent the full ensemble, given the relative strengths and weaknesses of individual ESMs, and if the selection was done with specific regions in mind it would be likely different.

**Specific Comments**

L34: (CMIP; Eyring et al 2016) > in LaTeX, (CMIP; \citealp{Eyring})

L35-36: Same as L34

L38-39: \citep[e.g.][]{X,Y,Z}

L63-64: Same as L38-39

L73-74: Same as L38-39

Thank you for highlighting the above copyediting. We have corrected them.

L135: Table1?

Thank you, we've corrected this.

Table 2: ECS values are available for your missing models:

- CMCC-CM2-SR5: Values reported in the IPCC's Assessment Report 6 Working Group I Chapter 7 Supplementary Material (The Earth's energy budget, climate feedbacks, and climate sensitivity) Table 7.SM.5.
- EC-Earth3-Veg-LR and FGOALS-g3: https://github.com/mzelinka/cmip56_forcing_feedback_ecs

- NorESM2-MM: Seland, Ø., Bentsen, M., Gra?, L., Olivié, D., Toniazzo, T., Gjermundsen, A., Debernard, J., Gupta, A., He, Y., Kirkevåg, A., Schwinger, J., Tjiputra, J., Aas, K., Bethke, I., Fan, Y., Griesfeller, J., Grini, A., Guo, C., Ilicak, M., and Michael, S.: The Norwegian Earth System Model, NorESM2 – Evaluation of the CMIP6 DECK and historical simulations, https://doi.org/10.5194/gmd-2019- 378, 2020a.

Thank you! We have sourced these values, added the citation to Zelinka (as this covers all 4 models)  and re-run our analyses to include these models as candidates for selection (they were always included in the set of models making up the full data in each experiment). The final subsets for each experiment were unchanged.

L159-161: Is it fair to compare individual realizations to an ensemble average for things like interannual standard deviation? Additionally, how do you handle the fact ensemble spread in precip. is much larger than ensemble spread in temperature for many regions?

We believe it is fair, since standard deviation is first computed on each individual ensemble member available and then these standard deviation values are averaged across the ensemble. We have further clarified this order of operations in the manuscript (L175-183).  Doing the ensemble average first and then taking the standard deviation of the resulting time series would have issues along the lines you highlight, that an ensemble average from a size of 1 would potentially have much higher IASD than an ensemble average from a size of 50.  As we do it in the manuscript, we feel it is fair (although of course larger ensembles give a more robust estimate of ensemble average IASD). As for different spreads, all PCA analyses in this work are conducted on centered and scaled variables, as noted in the legend of figure 2.

Figure 1: Can the side-by-side panels be on the same y axis scale? Additionally, the figure titles are identical, is this intentional?

Thank you for this suggestion and for catching our typo in the titles. We have adjusted the y-axes to be consistent and corrected the title.

L206: Can you give a sense of what the sign of the PC represents? Could you show the full ensemble variance spatially? That might help with the interpretation of the EOF.

Thank you for raising these questions. We have attempted to add text clarifying this (L238-249).

But briefly: with eigenvectors, the sign of the total vector (as represented by all six maps together in each row) is irrelevant (if v is an eigenvector, so is -v with the same eigenvalue). For the components within each eigenvector, eg the temperature IASD map being quite red in PC2 compared to the lighter overall blue mid century anomaly map in PC2, the sign is also not directly meaningful because the variables are centered and scaled. The critical piece in understanding the EOFs is tracking dominance, for lack of a technical term, in each PC. One can interpret PC1 as showing temperature being the driving factor for that fraction of total variance, PC2 primarily as showing the importance of internal variability in explaining that portion of total variance, precipitation trends being mostly what is explaining the portion of total variance explained by PC3, etc. These are good sanity checks (e.g. temperature is the most important thing in explaining the biggest fraction of total variance in climate data, we probably expect that) but interpretation is not as relevant as the mathematical fact that PCA results in an orthogonal coordinate system that maximizes total variance.

We did check the plots of total variance across models and scenarios for each of the six indices in each region, plotted here, but we are not sure what value it would add to the paper to include. Note that because this is on the raw data rather than the centered and scaled data the PCA was performed on, each index has its own units and ranges and so of course does the variance of each index.

[Figure]

L215: Can you elaborate on "strikingly similar"?

We have added additional text (L254-257).

L222: What features?

Thank you, we have clarified (L265). We simply mean that having dependent models with, for example, shared cloud physics in the full data means that the full data has a bias towards that physical representation of clouds compared to other representations present in fewer models.

L235: Can you comment on what is happening with MRI is PC2?
Figure 3 and Figure 4: These scatters are not so legible, and I'm not sure what is to be gained from scattering each PC against the others? Would scatters of each PC against total variance illustrate the message?

We agree that the figures are dense and the message we hope to convey with them is not well-illustrated currently. We have clarified in the text that the point of these figures is less the particular values plotted, and more that the cloud of points together represents a five-dimensional surface we are intelligently selecting models to span from those currently available (L278-291).

These are not quite scatters against the PCs, they are projections of each model's index data into the vector space defined by the PCs, i.e. plotting the cij values from L203. We have clarified this in the text as well (L281-286). This is the 5 dimensional cloud of points that our metric selects spanning representatives of.  We aren't sure what you mean by scattering a principal component against total variance. Figure 1 does plot the fraction of total variance that each principal component explains.

L258-260: I think I am misunderstanding. Is the idea to have the subset sit at the center of the distribution? Or to cover the spread? I drew a scenario that I think illustrates my confusion about the summary metric.

We have clarified the goal of our selection criteria in the manuscript (L112-122). It is closer to the latter.

[Figure]

Smaller summary metric

Larger summary metric

L305: stray box in the equation.

Thank you, we have corrected this.

Figure 5: Why do you think you lose temperature agreement in so many regions in Experiment 2?

It is likely reflecting that the full data is different in Experiment 1 than it is in Experiment 2, and therefore so are the respective distributions of ECS for each Experiment's full data. So effectively, the baseline data against which our subsets get compared are different. Therefore the effect of a constrained ECS distribution is just different relative to the ECS distribution of Experiment 1's full data than to Experiment 2's full data.

Figure 6: My read here is that in all the regions you flag, model uncertainty is always under-represented by the subset (due to the constraint on ECS you impose) and the partition

between scenario uncertainty and interannual variability in the subset approaches the full ensembles over time, but scenario uncertainty of the subset is always under that of the full ensemble in 2040. Is this to be expected? Do we see some cases where the subset has more scenario uncertainty than the full ensemble early in the record? Though model uncertainty is shifted with respect to the full ensemble it seems to evolve through time in a similar way in most cases. Isn't that more important than just a RMSE < 0.1?

Thank you for highlighting this. We have clarified in the text that the RMSE threshold is primarily a way to manage inspection of so many time series (L406-412 and L446-448), and we have further emphasized the point you raise - that matching the overall evolution in time is more critical than exact agreement. We actually do not think that scenario uncertainty is less at 2040 than in the full ensemble. Scenario uncertainty is given by the difference between the solid and dashed curves which behave in most cases indistinguishably from the green wedge at 2040, and always consistently with the expectation that scenario uncertainty would be the lesser source of uncertainty, often close to negligible, at that time. We did compare the scenario uncertainty fractions for the subset and full data and in most regions from 2035-2045, they differ by only 1-2% of total variance attributed to scenario uncertainty. I.e. the subset might say 7.5% of total variance in a region is due to scenario uncertainty in 2040 where the full data indicates 6%. As you point out in your comment, from a wholistic perspective those values are saying the same thing: in 2040, a very small fraction of total variance in every region is due to scenario uncertainty, which we expect because the 2030s-2040s is when the ScenarioMIP experiments considered in this work (SSP126, 245, 370, 585) begin to diverge meaningfully.

Figure 7: Again, I see a difference between cases where the subset is shifted down w.r.t. the full ensemble partition (e.g., Experiment 2 ESB) and cases where the partition is fundamentally different in time (e.g., Experiment 2 EAS). Have you investigated why this might be in more detail?

We feel this is outside the scope of this work as the focus of our work is proposing a methodology. We are proposing a method that has some good qualities in a global sense, so we are not expecting perfect performance everywhere. We emphasize in the introduction and conclusions that the specific choices around spatial regions and discretization considered in this work are driven by our experience with models that require global coverage. We often accept issues in specific regions in these applications in exchange for the coverage needed. We also have added text regarding this to the results (L450-455).

**Reviewer 3:**

Review of esd-2023-41

**Title: Uncertainty-informed selection of CMIP6 Earth System Model subsets for use in multisectoral and impact models**
 Authors: Abigail Snyder, Noah Prime, Claudia Tebaldi, Kalyn Dorheim

***Overall Recommendation: Accept***

This study, submitted to Earth System Dynamics, highlights a new approach to select subsets of CMIP6 GCMs for use in multi-sectoral and impacts modeling. The approach is novel and attempts to capture multiple sources of uncertainty in the subset and does have potential utility for multiple applications beyond what was mentioned in the manuscript. This is the second time I have reviewed this manuscript and I'm pleased that the authors addressed my previous comments. As my remaining comments are minor issues, I recommend moving forward with publication. I look forward to seeing the revised manuscript in print.

Thank you for your feedback. Our responses are in blue and **all line numbers referenced refer to line numbers in the new track changes manuscript.**

Minor Comments:

Line 135 – There's a typo where Tables 1, 2, and 3 are all mentioned in this line. I believe Table 2 was what the author's meant to refer too.

Thank you, we've corrected this. A victim of moving things around with track changes across platforms.

Lines 138-142: "Models for which we... (more details in Section 2.2)." – How do you know the distribution of ECS is preserved when those without an ECS value in literature are removed? Presumably, these models have an ECS value, it is simply that the ECS values for thos GCMs have not been assessed by other literature.

This is a very good point.

Each subset considered is a set of 5 models with available ECS distributions that satisfy the IPCC distribution. You are correct that other subsets including the models we did not previously have ECS values for do exist that satisfy this. We have attempted to further emphasize in the conclusions that this is a method-focused work that scientists can adapt for information they have access to and prioritize, rather than a claim that our final subsets are in anyway 'the best'. They are simply the most numerically optimal at this particular task with the particular choices outlined in table 1.

Incidentally, the other two reviewers did provide citable ECS values for the 4 models we did not have previously. We re-ran our analysis to include subsets with these as candidates (considering a total of 150 subsets compared to the previous 72 for experiment 1), and the resulting selected subset in both Experiment 1 and Experiment 2 did not change. Two of the four newly added models (CMCC-CM2-SR5 and FGOALS-g3) are dependent with models we previously considered for selection in Experiment 1 and the projections between these

dependent pairs of models are quite similar in the eigenvector space (which always was based on all available models, independent of ECS values), so it is in retrospect not surprising that new subsets with those two models were not selected to span the space efficiently. For subsets containing the other two of the four models  (EC-Earth3-Veg-LR and NorESM2-MM), it is more down to chance that subsets containing them were not selected.

Line 189 – Like the comment for Line 135, there's a typo where Tables 1, 2, and 3 are mentioned in the same line.

Thank you, we've corrected this.

Lines 199-201: "Based on this figure...flexibility of this method." – The authors did respond to my question with respect to the number of chosen eigenvectors. I suggest including the response here rather than only in the response to the reviewers.

Thank you, we have done so (L228-232).